# An epigenome atlas of neural progenitors within the embryonic mouse forebrain

Christopher T. Rhodes[1], Joyce J. Thompson[2], Apratim Mitra [3], Dhanya Asokumar [1,2], Dongjin R. Lee[1], Daniel J. Lee[1,2], Yajun Zhang[1], Eva Jason[3], Ryan K. Dale[3], Pedro P. Rocha [2,4] & Timothy J. Petros [1✉]

A comprehensive characterization of epigenomic organization in the embryonic mouse forebrain will enhance our understanding of neurodevelopment and provide insight into mechanisms of neurological disease. Here we collected single-cell chromatin accessibility profiles from four distinct neurogenic regions of the embryonic mouse forebrain using single nuclei ATAC-Seq (snATAC-Seq). We identified thousands of differentially accessible peaks, many restricted to distinct progenitor cell types or brain regions. We integrated snATAC-Seq and single cell transcriptome data to characterize changes of chromatin accessibility at enhancers and promoters with associated transcript abundance. Multi-modal integration of histone modifications (CUT&Tag and CUT&RUN), promoter-enhancer interactions (Capture-C) and high-order chromatin structure (Hi-C) extended these initial observations. This dataset reveals a diverse chromatin landscape with region-specific regulatory mechanisms and genomic interactions in distinct neurogenic regions of the embryonic mouse brain and represents an extensive public resource of a 'ground truth' epigenomic landscape at this critical stage of neurogenesis.

[1] Unit on Cellular and Molecular Neurodevelopment, Eunice Kennedy Shriver National Institute of Child Health and Human Development (NICHD), National Institutes of Health (NIH), Bethesda, MD 20892, USA. [2] Unit on Genome Structure and Regulation, Eunice Kennedy Shriver National Institute of Child Health and Human Development (NICHD), National Institutes of Health (NIH), Bethesda, MD 20892, USA. [3] Bioinformatics and Scientific Programming Core, Eunice Kennedy Shriver National Institute of Child Health and Human Development (NICHD), National Institutes of Health (NIH), Bethesda, MD 20892, USA. [4] National Cancer Institute (NCI), NIH, Bethesda, MD 20982, USA. ✉email: tim.petros@nih.gov

nhibitory GABAergic interneurons are a heterogeneous cell population that can be classified based on electrophysiological properties, morphologies, synaptic connectivity, neurochemical markers, and transcriptomes[1–3]. In the forebrain, GABAergic neurons are born from transient embryonic structures in the ventral telencephalon known as the medial, caudal, and lateral ganglionic eminences (MGE, CGE, and LGE, respectively), whereas glutamatergic projection neurons arise from the dorsal telencephalon. The MGE and CGE (and adjacent preoptic area) generate nearly all cortical and hippocampal interneurons, with each region generating almost entirely distinct, non-overlapping interneuron subtypes[4–6].

The embryonic brain contains two primary classes of neural progenitors: multipotent self-renewing apical progenitors (APs, also known as radial glia cells) located in the ventricular zone (VZ) and basal progenitors (BPs) that undergo neurogenic divisions within the subventricular zone (SVZ)[7]. Both APs and BPs give rise to postmitotic immature neurons (Ns) within the GEs that migrate tangentially to populate the telencephalon. Recent evidence demonstrates that initial interneuron subtype fate is specified within the GEs as interneuron progenitors exit the cell cycle[8–11]. It is well established that changes in a cell's epigenetic landscape alter cell fate decisions throughout normal development[12,13] and can be associated with neurodevelopmental disorders[14–16]. In fact, many neurological and psychiatric disease-associated genes are expressed during embryonic development[17,18] and are enriched specifically in APs and immature interneurons[19,20]. Furthermore, many neurological disorders have been linked directly to polymorphisms in enhancer regions[21,22], and GWAS indicates that >90% of disease-associated single nucleotide polymorphisms (SNPs) are located outside of coding regions[23]. Thus, a thorough characterization of the epigenomic landscape during neurogenesis is necessary to understand normal development and potential disease etiologies.

Using a single nuclei assay for transposase accessible chromatin followed by sequencing (snATAC-Seq)[24], we characterized the chromatin accessibility of cells during the transition from progenitors to lineage-restricted neurons within the GEs and dorsal telencephalon of the embryonic mouse brain. We identified differentially accessible peaks (DA peaks) enriched in specific brain regions and/or distinct progenitor cell types. Among chromatin accessibility profiles, individual loci smoothly transition from open to closed chromatin (or vice versa) during the initial stages of neurogenesis. We validated our snATAC-Seq and single-cell RNA sequencing (scRNA-Seq) observations with orthogonal epigenomic methods. Genome-wide histone modification profiles associated with promoters (H3K4me3), active enhancers (H3K27ac), and gene repression (H3K27me3) were highly concordant with our snATAC-Seq profiles showing spatially restricted enrichment patterns. Our single-cell derived gene-enhancer models largely agreed with direct observations of promoter–enhancer interactions by Capture-C and higher-order chromatin domains by Hi-C. These data are available as a UCSC Genome Browser track hub, providing an important new resource for the field to explore spatial differences in the chromatin landscape of distinct neuronal progenitors within the embryonic mouse forebrain.

## Results

**Identifying chromatin accessibility profiles in the embryonic mouse forebrain.** To ascertain the chromatin accessibility landscape of differentiating neurons, we dissected the MGE, CGE, LGE, and cortex from wild-type mice at embryonic day 12.5 (E12.5) when cells in the GEs are undergoing neurogenesis[4] and processed single nuclei on the 10X Genomics platform (Fig. 1a). Since cortical neurogenesis occurs later, we also harvested E14.5

dorsal cortex to compare both temporally (E12.5 GEs vs. E12.5 cortex) and neurogenically (E12.5 GEs vs. E14.5 cortex) matched dorsal and ventral forebrain. Sequencing libraries contained 39,253 single nuclei, with 10,310 from MGE, 8543 from CGE, 11,346 from LGE, and 9054 from the cortex. Libraries were aggregated, downsampled to equal numbers of median fragments per nuclei, and normalized by latent semantic analysis (LSA) before peak calling, construction of cell-by-peak count matrices, and integration of different samples (Supplementary Fig. 1a–h).

Using uniform manifold approximation and projection (UMAP), nuclei were segregated largely by tissue region (Fig. 1b). The smart local moving (SLM) algorithm[25] detected 27 clusters, of which three non-neuronal clusters were removed to retain 96.8% of nuclei in 24 clusters (Fig. 1c). Cell types were assigned by inspecting promoter accessibility (PA) (defined as the sum of reads mapping within −2000 bp of a TSS) of canonical cell type markers and were further refined by transferring cell type assignments from droplet-based scRNA-Seq data of E12.5 embryos (Fig. 1d and Supplementary Fig. 1i–n) to the snATAC-seq dataset. PA for markers of interneuron and excitatory glutamatergic pyramidal cell maturation segregated clusters into mitotic APs and BPs, and postmitotic Ns (Fig. 1d and Supplementary Fig. 1o–q). MGE and most CGE and LGE nuclei displayed accessible chromatin at GABAergic neuron markers, while virtually all cortical nuclei have accessible chromatin at markers of glutamatergic neurons (Supplementary Figs. 1o, 2). A group of LGE and CGE nuclei displayed accessibility profiles more similar to glutamatergic neuron markers and were labeled as a "mixed" neuron population (Supplementary Figs. 1o, 2). This was likely contamination from the pallial-subpallial boundary (PSB), a region that gives rise to cells located in the piriform cortex, claustrum, and amygdala[26,27].

To quantify temporal differentiation programs, a minimum spanning tree was constructed in Monocle3[28,29] (Fig. 1e). The progression along pseudotime largely recapitulated known maturation markers, from Nes + and Ccnd2 + cycling progenitors to Dcx + and Rbfox3 + postmitotic neurons (Fig. 1f–i). Additionally, region-restricted genes such as Nkx2-1 and Lhx6 in the MGE, and Neurod6 in the cortex displayed open accessibility profiles restricted to these regions (Fig. 1j–m). We examined pseudobulk ATAC read pileups within each cluster for regionally restricted genes for the MGE (Nkx2-1) and cortex (Neurod6) (Fig. 1n, o) and for two pan-neuronal maturation markers for APs (Nes) and BPs/Ns (Dcx) (Fig. 1p, q). High signal strength for Nkx2-1 and Neurod6 reads were restricted to the MGE and cortex/mixed clusters, respectively. As expected, Nes reads were enriched in AP clusters with diminished signals in BP and N clusters, whereas Dcx displayed the inverse low-AP to high-N accessibility profile. Notably, strong accessibility was detected in the second intron of Nes which contains a known enhancer[30] (Fig. 1p). These observations were in agreement with ENCODE H3K4me3 ChIP-Seq data from E12.5 mouse forebrain (Fig. 1n–q).

**Differentially accessible peak profiles during neurogenesis in the embryonic mouse forebrain.** When comparing differentially accessible (DA) peaks among all possible peaks (intergenic peaks and those in promoters/gene bodies), we sought to detect cluster and cell type-specific markers, detecting a total of 30,046 DA peaks (FDR ≤ 0.05, average log(fold change) > 0) across all clusters (Fig. 2a, Supplementary Fig. 3a, and Supplementary Data 1). These DA peaks represent accessible genomic loci that are potentially unique to specific cell types. To characterize DA peak profiles across clusters, we asked whether the genomic coordinates bounding DA peaks of one cluster had reads in any

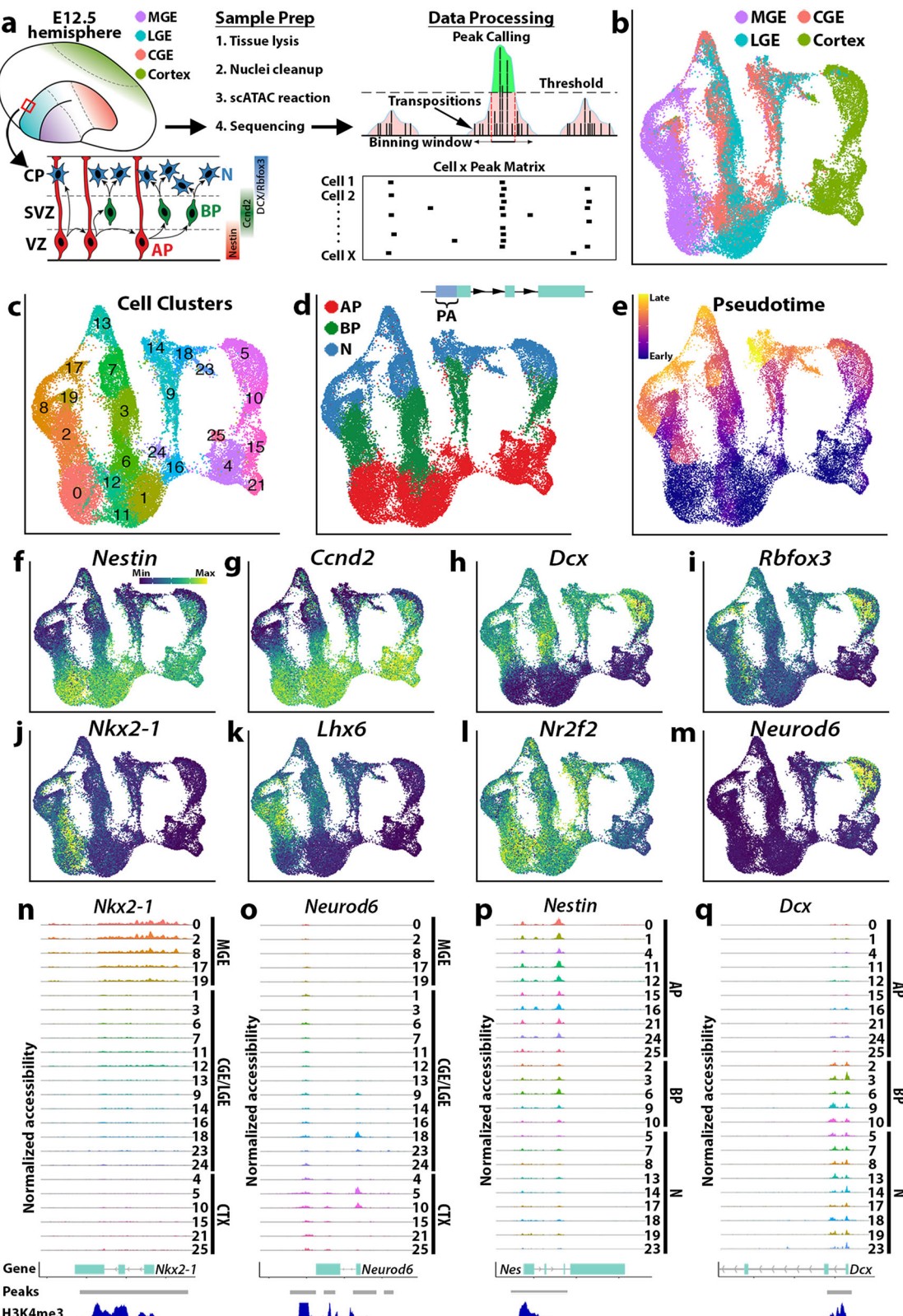

**Fig. 1 Chromatin accessibility in the mouse embryonic forebrain is cell type and state-specific. a** Schematic of snATAC-Seq workflow and neurogenic cell types: apical progenitors (APs), basal progenitors (BPs), and posmitotic neurons (Ns). **b–e** UMAP visualization of single nuclei clustered by brain region (**b**), SLM (**c**), neurogenic cell type (**d**), and pseudotime (**e**). In **d**, promoter accessibility (PA) representing reads mapping within 2 kb upstream of TSSs. **f–m** PA scores for genes enriched in specific neurogenic cell types (**f–i**) or distinct brain regions (**j–m**). **n–q** Aggregated reads per SLM cluster. *Nkx2-1* (MGE), *Neurod6* (cortex), *Nes* (APs), and *Dcx* (BPs/Ns), arranged by either brain region (**n, o**) or neurogenic cell type (**p, q**). The y-axis range for chromatin accessibility tracks are normalized to the maximum reads per gene. Peaks: differentially accessible peak coordinates, H3K4me3: H3K4me3 signal from E12.5 forebrain ENCODE ChIP-Seq data.

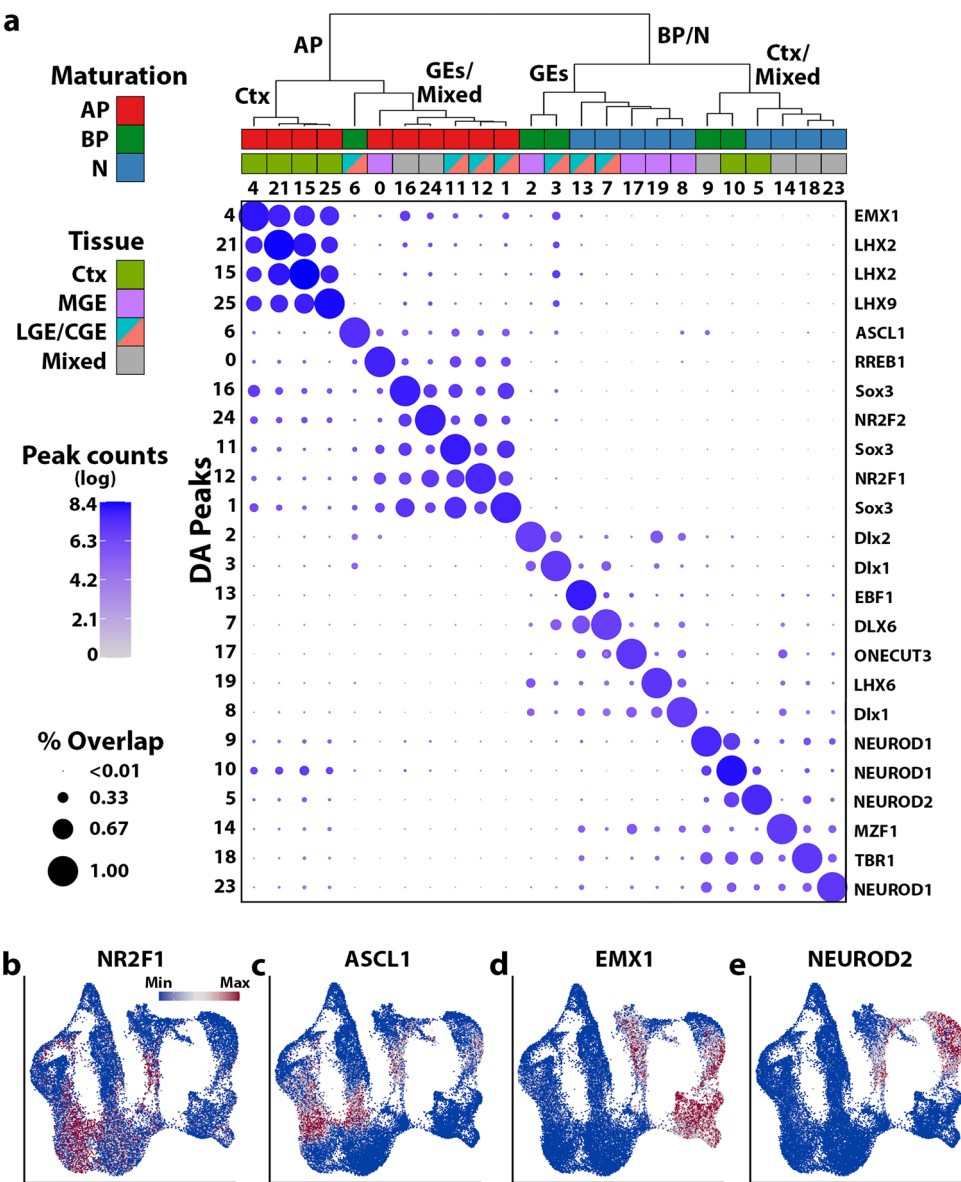

**Fig. 2 Differentially accessible peaks are cluster and lineage-specific in the mouse embryonic forebrain. a** Embryonic snATAC-Seq dot plot of differentially accessible peaks (DA peaks) for each cluster. Dot diameter indicates the percent of DA peaks from one cluster (column cluster labels) which are detectable in any other cluster (row cluster labels). Color intensity represents the total DA peak count per cluster. Hierarchical clustering was performed using correlation distance and average linkage. Names of representative cluster-enriched transcription factor binding motifs in DA peaks are listed to the right of the dot plot. **b–e** UMAP plots of nuclei colored by ChromVAR global motif deviations (Z-scores) for NR2F1, ASCL1, EMX1, and NEUROD2.

other cluster. If there were reads in a DA peak from one cluster in another cluster, this peak was considered overlapping between the clusters. No minimum threshold for overlapping peak counts was used before calculating the percentage of overlapping DA peaks from one cluster compared to all peaks from the same cluster (Fig. 2a, "% Overlap"). We also counted the number of DA peaks from each cluster to assess if there were differences in the number of DA peaks per cluster (Fig. 2a, "Peak counts").

Unsupervised hierarchical clustering (HC) of DA peak profiles created a dendrogram that segregated clusters initially by maturation state (AP, BP, and N) and secondarily by tissue origin. Since the overall profiles of LGE and CGE nuclei were very similar, these two regions were labeled 'LGE/CGE' for this analysis. The 'mixed' neuron population (Supplementary Fig. 1o) was also left as an individual group for this analysis. The dendrogram generated by HC is very similar to cluster

relationships in LSA/UMAP space, which is encouraging since different features were used in each analysis (HC: DA peaks in cluster pairs, LSA: all peaks across libraries). As expected, dot plot positions containing both high DA peaks counts and high percent overlap were almost exclusively grouped along the diagonal, while positions with limited numbers of DA peaks or low between-cluster peak overlap populated the off-diagonals, indicating high specificity of DA peaks to specific clusters (Fig. 2a). The cell type- and brain region-specific DA peak profiles are consistent with previous models of chromatin reorganization during cellular differentiation[31,32].

We also visualized binarized peak signals (i.e., "open" or "closed" regions) per cluster using a heatmap and again observed high peak signals primarily along the diagonal (Supplementary Fig. 3a). Intriguingly, the mean number of DA peaks decreased as maturation progressed, with a significant decrease in DA peaks in

BP and N nuclei compared to APs in each tissue (Supplementary Fig. 3c–d). Despite this decrease, DA peak profiles became more distinct as maturation progressed, as indicated by the low between-cluster peak overlap along the dot plot off-diagonal (Fig. 2a and Supplementary Fig. 3a). The decrease in global accessibility over time is consistent with previous observations during cellular differentiation[33]. We observed a greater number of DA peaks in all maturation stages in the cortex compared to GEs (Supplementary Fig. 3e–f).

Regions of accessible chromatin are enriched for transcription factor (TF) binding motifs that often play essential roles in driving cell specification. To characterize region-enriched TF motifs, we performed motif analysis on DNA sequences within DA peaks using the JASPAR CORE[34] vertebrates collection. Motif analysis detected TF motifs that have lineage- and tissue-specific roles during interneuron neurogenesis (Fig. 2a, Supplementary Fig. 3b, and Supplementary Data 2). Further, among motifs enriched in specific clusters that also had a corresponding differentially accessible promoter, the top five matches contain one or more TF motifs with known spatial or temporal expression profiles correlating with the cell cluster (Fig. 2b and Supplementary Data 3). For example, UMAP visualization of motif enrichment by chromVAR[35] for NR2F1 (MGE and CGE-specific)[36], ASCL1 (GABAergic BP-enriched), EMX1 (cortical progenitor AP/BP-specific), and NEUROD2 (cortical postmitotic N-specific) motifs demonstrated the expected neuronal lineage progression and/or region-restricted patterns (Fig. 2b–e). We further quantified genome-wide DA peak distributions within annotated gene regions and found the majority of DA peaks were constrained to the promoter and intronic regions of gene bodies, and distal intergenic loci (Supplementary Fig. 3g). These findings indicate that DA peaks are specific to brain region and cell-state, and importantly, that DA peaks contain lineage-specific TF motifs that may regulate cell fate decisions differentiation.

**Candidate cis-regulatory elements are dynamic and cell state-dependent in the embryonic mouse forebrain.** The global decrease of DA peak numbers from cycling neural progenitors to postmitotic immature neurons prompted us to examine changes to DA peaks during maturation and lineage commitment. For this and all future analyses, we removed the 'mixed' population to characterize only LGE and CGE-derived GABAergic cells. The Monocle3 extension Cicero[37] detects interactions or "connections" between any two genomic loci and then assigns a co-accessibility score between the two sites, thereby calculating the proportion of nuclei containing a given co-accessibility interaction within a population. We quantified such interactions within all nuclei and detected 92,414 connections that had a co-accessibility score equal to or greater than 0.25, meaning a given locus-locus interaction is detectable in >25% of all nuclei in the defined population (henceforth "Cicero connections").

To explore co-accessibility changes during neurogenesis, we used a heatmap to visualize Cicero connections in which at least one of the two interacting loci was a DA peak, representing potential interactions between DA peaks and putative cis-regulatory elements (cREs). Nuclei were divided into 10 bins of equal pseudotime intervals along the Y-axis, with individual DA peaks grouped via hierarchical clustering along the X-axis. The heatmap color represents the proportion of nuclei with a given DA peak (interacting via a Cicero connection) with a co-accessibility score >0.25 (Fig. 3a). Nearly half of these Cicero connections were enriched in AP nuclei, which is consistent with an overall decrease in accessibility as development progresses (Supplementary Fig. 3c–d). As chromatin regions with dynamic accessibility are associated with gene regulation during neural

stem cell activation[32], we hypothesized that Cicero connections enriched in immature neurons encode lineage-specific cREs that may play a role in neuronal lineage commitment.

To identify candidate cREs at specific genes, we examined Cicero connections within 0.5 Mb windows of gene TSSs after filtering for co-accessibility scores >0.25. We identified Cicero connections in which only one of the two interacting loci overlapped a TSS, representing potential interactions between a TSS and putative cREs (henceforth "TSS-cRE connections"). All Cicero connections within this 0.5 Mb window are visualized as orange arcs whereas TSS-cRE connections for a selected gene are highlighted in purple (Fig. 3b, c). To characterize Cicero connections that were spatially restricted, we downsampled tissues to equal nuclei numbers and detected 91,904, 76,858, 89,366, and 148,942 Cicero connections with co-accessibility scores >0.25 in MGE, LGE, CGE, and cortex, respectively. *Lhx6* and *Neurod6* Cicero connections are restricted to the MGE and cortex, respectively, and TSS-cRE connections for *Lhx6* and *Neurod6* are virtually exclusive to these regions (Fig. 3b). *Nr2f2* (*CoupTF-II*) is a marker for APs in the VZ of the CGE and caudal MGE[38] whereas *Sp8* is a marker for LGE progenitors that is excluded from the MGE[39]; both genes displayed highest co-accessibility scores and TSS-cRE connections counts in the expected regions (Fig. 3b). Overall, tissue-specific TSS-cRE connectivity patterns were similar to regionally restricted gene expression patterns that are critical to neuronal development.

To characterize Cicero connections that varied between neurogenic cell types, we downsampled to equal nuclei numbers and detected 89,312, 88,980, and 110,362 Cicero connections with co-accessibility scores >0.25 in AP, BP, and N nuclei, respectively. Among these Cicero connections, postmitotic genes *Lhx6* and *Neurod6* had their highest co-accessibility scores and TSS-cRE connections in BP and N nuclei (Fig. 3c). Progenitor-enriched genes *Nr2f2* and *Sp8* had their highest co-accessibility scores and TSS-cRE connections in APs, with decreased connections and co-accessibility scores in Ns. The pan-AP marker nestin (*Nes*) had its highest co-accessibility scores and TSS-cRE connections in APs throughout all regions compared to BPs and Ns. Conversely, the postmitotic GABAergic marker *Gad1* had its highest co-accessibility scores and connection counts in BPs and Ns. As with the regional specificity of TSS-cRE connections, the temporal connection patterns largely recapitulate known gene expression patterns as neurons mature. In sum, cREs likely interact with DA peaks and TSSs to regulate genes in regionally and temporally restricted patterns, and the co-accessibility patterns of TSS-cREs connections closely resemble known spatial and temporal expression patterns in the embryonic forebrain.

**Integrative analysis of chromatin accessibility and gene expression profiles in the embryonic mouse forebrain.** To enhance our understanding of the relationship between chromatin accessibility and gene expression during neurogenesis, we combined chromatin accessibility profiles from snATAC-Seq data with age and region-matched scRNA-Seq data. Integrating ATAC and RNA data involves quantifying ATAC reads in or near gene bodies by a Gene Activity Score (GAS) as a proxy for transcript abundance. After testing multiple GAS metrics for snATAC-Seq/scRNA-Seq integration, we defined GAS as the sum of all ATAC reads mapping to the promoter, first exon, and presumptive enhancers of a given gene because this GAS metric produced the highest concordance between ATAC and RNA assays (Supplementary Fig. 4). Following integration, the clustering of snATAC-Seq and scRNA-Seq cells were highly concordant, with the MGE and cortex integrated cells formed distinct clusters whereas the LGE and CGE cells were largely overlapping (Fig. 4a, b). The

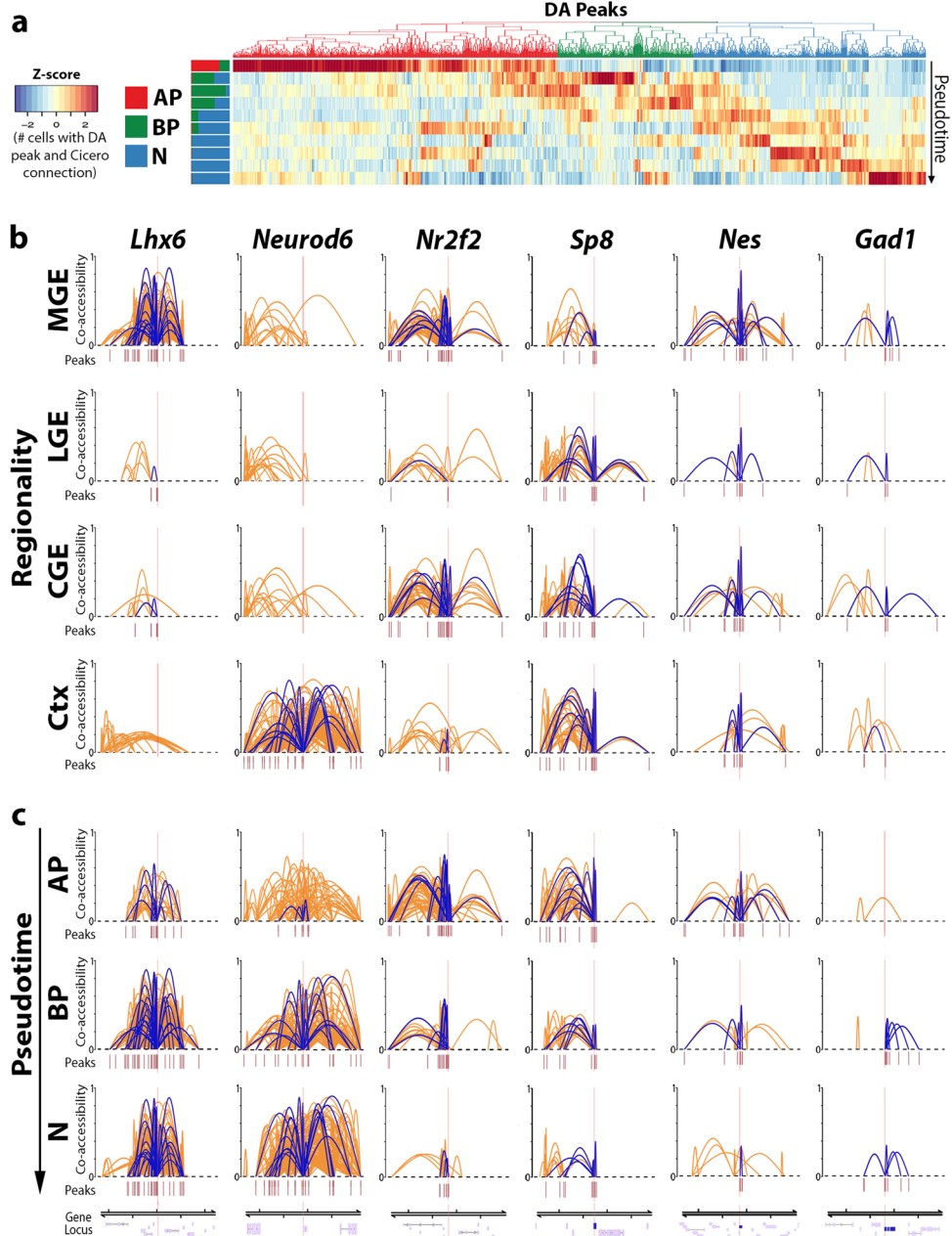

**Fig. 3 Detection of cis-regulatory elements within the developing mouse forebrain. a** Heatmap depicting DA peaks binned along pseudotime from embryonic snATAC-Seq nuclei. DA peaks were filtered to retain only peaks that had at least one Cicero peak-peak connection. Stacked bar plots to the left of the heatmap depict the proportion of AP, BP, and N nuclei per bin. **b**, **c** Cicero connections within 0.5 Mb window centered around TSS of *Lhx6* (postmitotic MGE marker), *Neurod6* (postmitotic cortex marker), *Nr2f2* (CGE progenitors), *Sp8* (LGE progenitors), *Nes* (pan-AP), and *Gad1* (postmitotic pan-GEs) broken down based on tissue (**b**) or neurogenic cell type (**c**). Cicero connections overlapping the TSS of selected genes are shown as purple arcs, all Cicero connections in the genomic regions are shown as orange arcs. Y-axis unit is the co-accessibility score. Only connections with co-accessibility scores greater than 0.25 are depicted. Peaks: snATAC peaks were used by Cicero to quantify peak-peak connections. Gene models are visualized at the bottom of each column with genes of interest highlighted.

Louvain algorithm detected 26 clusters (Fig. 4c) and Monocle3 assigned pseudotime (Fig. 4d) which largely recapitulated temporal and spatially restricted expression patterns expected in embryonic forebrain neurogenic regions (Fig. 4e–j and Supplementary Fig. 5).

Prior to integration, we refined our cRE predictions to detect presumptive enhancers by combining our Cicero TSS-cRE analysis with TSS-cRE predictions from the SnapATAC algorithm[40] (Supplementary Data 4). SnapATAC predictions link distal regulatory elements to target genes based on the transcript count of a gene and chromatin accessibility at peaks flanking the gene using gene expression as an input variable to predict the binarized chromatin state at peaks. Our rationale for using multiple algorithms was that TSS-cREs connections common to both methods likely improve cRE predictions. After combining TSS-cRE connections from both methods, we retained common cREs to generate a list of all detectable presumptive enhancers (Supplementary Fig. 6). We took the intersection of these loci and ENCODE H3K27ac ChIP-Seq peaks[41] from E12.5 and E14.5 forebrain, resulting in a set of 'high-confidence'

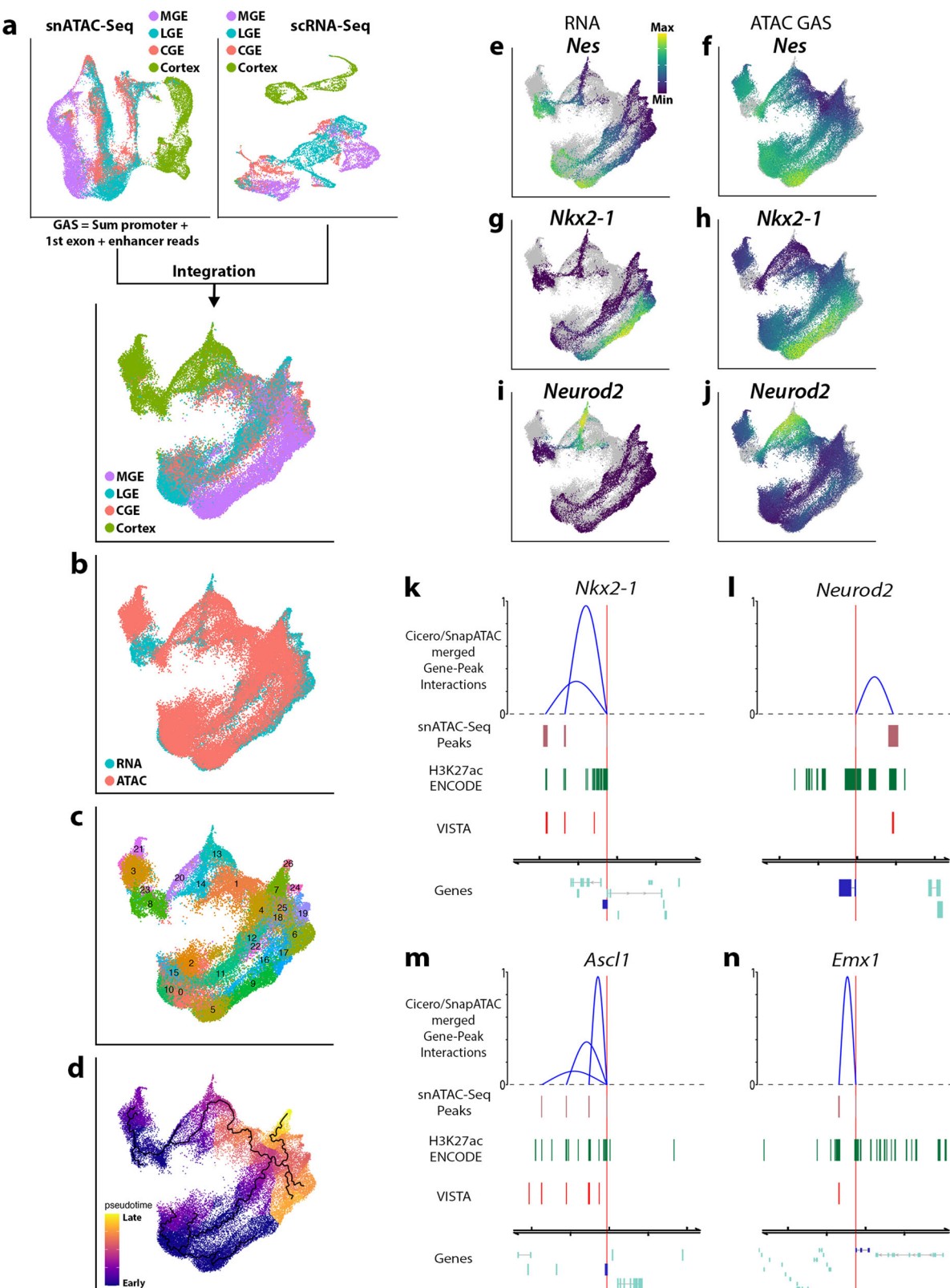

enhancer candidates (henceforth "presumptive enhancers") (Supplementary Data 5). We detected previously validated VISTA enhancers[42] interacting with genes in the MGE (hs704 and hs1538 regulating *Nkx2-1*), cortex (hs627 regulating *Neurod2*), GABAergic progenitors (hs967, hs998, hs1114, hs1354, and hs1540 regulating *Ascl1*) and glutamatergic progenitors (hs1025 regulating *Emx1*) (Fig. 4k–n and Supplementary Data 6).

To characterize the temporal expression and chromatin accessibility profiles during neurogenesis, we utilized a hierarchical clustering-based approach from the DEGReport package[43] to group RNA, GAS, and enhancer counts from our integrated scRNA-Seq/snATAC-Seq data (Fig. 5a–r). As not all genes had detectable snATAC-based GAS and/or enhancer counts, we selected ~300 of the most differentially expressed genes (DEGs)

**Fig. 4 Integrative analysis of snATAC-Seq and scRNA-Seq from embryonic mouse forebrain. a** Workflow depicting integration of embryonic snATAC-Seq (top left) and scRNA-Seq (top right) data. Bottom, UMAP plot showing integrated snATAC-Seq nuclei and scRNA-Seq cells colored by tissue. **b–d** UMAP visualization of integrated snATAC-Seq/scRNA-Seq data colored by assay (**b**), Louvain cluster (**c**), and pseudotime (**d**). **e–j** UMAP visualization of integrated scRNA-Seq cells and snATAC-sec nuclei colored by transcript counts or GAS for *Nes* (**e, f**), *Nkx2-1* (**g, h**), and *Neurod2* (**i, j**). Gray dots in the background represent cells/nuclei from other assays. **k–m** Genome browser tracks displaying enhancer predictions regulating *Nkx2-1* (**k**), *Neurod2* (**l**), *Ascl1* (**m**), and *Emx1* (**n**). Gene-Peak interactions are visualized in blue arcs, arc heights indicate the relative interaction scores between gene TSS (red line) and peaks. snATAC-Seq Peaks: displays co-accessible coordinates, H3K27ac ENCODE: Forebrain E12.5 H3K27ac peaks from ENCODE project, VISTA: enhancers from the VISTA Genome Browser project.

(from scRNA) among APs, BPs, and Ns that had corresponding GAS and enhancer counts. DEGReports hierarchical clustering uncovered five groups containing at least six genes (Supplementary Fig. 7a, b), of which over 85% of the genes fell into two categories: one cluster is consistent with high expression and accessibility profiles within progenitors (APs and BPs) that are downregulated in postmitotic neurons (156 DEGs, Fig. 5a), and another cluster with the complimentary profile (90 DEGs, Fig. 5b). Genes within these clusters displayed similar patterns of expression and chromatin accessibility over pseudotime. Visualization of representative early-expressed genes *Hes1* (Fig. 5c–f) and *Lmo1* (Fig. 5g–j) and later-expressed genes *Myt1l* (Fig. 5k–n) and *Lhx6* (Fig. 5o–r) demonstrate consistent trends for transcript, GAS, and enhancer counts over pseudotime. We quantified the number of high-confidence enhancers associated with upregulated DEGs as maturation progressed and identified 200 enhancers associated with DEGs that had a positive fold change from APs-to-BPs, 175 enhancers for BPs-to-Ns, and 269 enhancers from APs-to-Ns (Supplementary Fig. 7c). Likewise, we found 188 enhancers associated with DEGs that had a negative fold change from APs-to-BPs, 179 enhancers for BPs-to-Ns, and 320 enhancers from APs-to-Ns (Supplementary Fig. 7c). There is a gradual decrease in the ratio of the number of enhancers being activated versus the number being decommissioned as maturation progressed (Supplementary Fig. 7d), suggesting that a greater number of genes and associated enhancers are repressed as progenitors exit the cell cycle. Taken together, DEGs and associated enhancers exhibit reorganization during the transition from progenitors to lineage-committed postmitotic immature neurons.

We characterized the differentiation processes by visualizing matched heatmaps for RNA, GAS, and presumptive enhancer counts of highly variable genes (Fig. 5s–u). We selected transcript counts and corresponding GAS and enhancer counts for the top 500 most variable genes from the E12.5 integrated dataset, of which 210 had corresponding GAS and enhancer counts (Supplementary Fig. S7e–g). RNA, GAS, and enhancer count matrices for these genes were co-clustered using hierarchical clustering with a correlation distance metric and average linkage and visualized in matched heatmaps (Fig. 5s–u). Partitioning early, transitional and late expressing gene profiles with respect to pseudotime largely followed a continuous progression as cells matured from APs through Ns. Overall, there was a high similarity between (1) early and late gene expression patterns detected by degPatterns (Fig. 5a, b and Supplementary Fig. 7a, b) and (2) early and late expressing genes visualized in heatmaps (Fig. 5s–u), indicating distinct, dynamic expression and chromatin accessibility in APs, BPs, and Ns. By integrating multiple single-cell modalities, we characterized the chromatin accessibility and gene expression profile of distinct forebrain regions during neurogenesis.

**Histone modifications and higher-order chromatin organization reveal region-specific chromatin states in the embryonic mouse forebrain.** Predicting enhancers from snATAC data has

enormous potential for mapping regulatory elements in heterogeneous cell populations. To validate some of these predictions, we carried out two additional sets of experiments. First, we performed CUT&RUN[44] and CUT&Tag[45] on E12.5 MGE, LGE, CGE, and cortex to detect histone modifications associated with active/poised promoters (H3K4me3), active enhancers (H3K27ac), and repressed genes (H3K27me3)[46–48]. Most genes with spatially restricted mRNA and promoter accessibility profiles contained corresponding H3K4me3 peaks whereas spatially repressed genes were enriched with H3K27me3 (Fig. 6a). More globally, we observed that ~70% of ATAC peaks at promoters overlapped with H3K4me3 marks in all brain regions (Fig. 6b).

To further identify candidate enhancers, we combined our Cicero analysis with H3K27ac enrichment. There was less overlap between ATAC peaks and H3K27ac marks (29.2–45.1%), as expected due to the weaker correlation between accessibility and H3K27ac marks throughout the genome. However, if we restricted analysis to ATAC peaks with a Cicero connection to a promoter (indicative of possible enhancers), then the percent overlap of ATAC peaks with H3K27ac marks increased considerably (54.9–68.1%) (Fig. 6b).

We observed region-specific colocalization between ATAC peaks, Cicero connections, and H3K27ac marks at many genes, some of which represent candidate enhancers. For example, there are VISTA enhancers downstream of the GE-enriched *Ascl1*, with one site (hs1540) showing co-accessibility in nearly all nuclei (Fig. 6c). However, none of these VISTA reporters displayed the expected GE-restricted *Ascl1* expression pattern (https://enhancer.lbl.gov) nor contained GE-enriched H3K27ac marks (Fig. 6c). Instead, we identified two other regions with Cicero interactions with GE-enriched H3K27ac marks compared to the cortex (Fig. 6c, gray bars). We identified similar loci near *Lhx6* and *Neurog2* with enriched H3K27ac marks specifically in the MGE and cortex, respectively, representing potentially unexplored cREs (Fig. 6d, e).

We performed Hi-C to characterize chromatin structure genome-wide (Fig. 7a) and Capture-C to directly quantify promoter interactions at ~50 genes with tissue-specific expression patterns (Fig. 7b and Supplementary Data 7). At the *Nkx2-1* locus, Hi-C data revealed the formation of an MGE-specific chromatin domain compared to other brain regions (Fig. 7a). Capture-C confirmed these distinct interactions, with the *Nkx2-1* promoter interacting directly with a region near the *Mbip* gene specifically in the MGE (Fig. 7b). Notably, *Mbip* expression is also restricted to the MGE during development[49]. Conversely, interactions between the *Nkx2-1* promoter and the *Nkx2-9* and *Pax9* loci (genes not expressed in the embryonic forebrain) were specifically detected in the LGE, CGE, and cortex (Fig. 7b). While the exact nature of these interactions is unclear (promoter–enhancer, promoter–promoter, etc.), the formation of region-specific chromatin domains is not observable from the other assays, as the snATAC and histone modifications at the *Mbip*, *Nkx2-9*, and *Pax9* locus are quite similar between the different brain regions (Fig. 7c).

Both Hi-C and Capture-C data identified a direct interaction between the *Nr2f1* promoter and an intron within *2210408I21Rik*

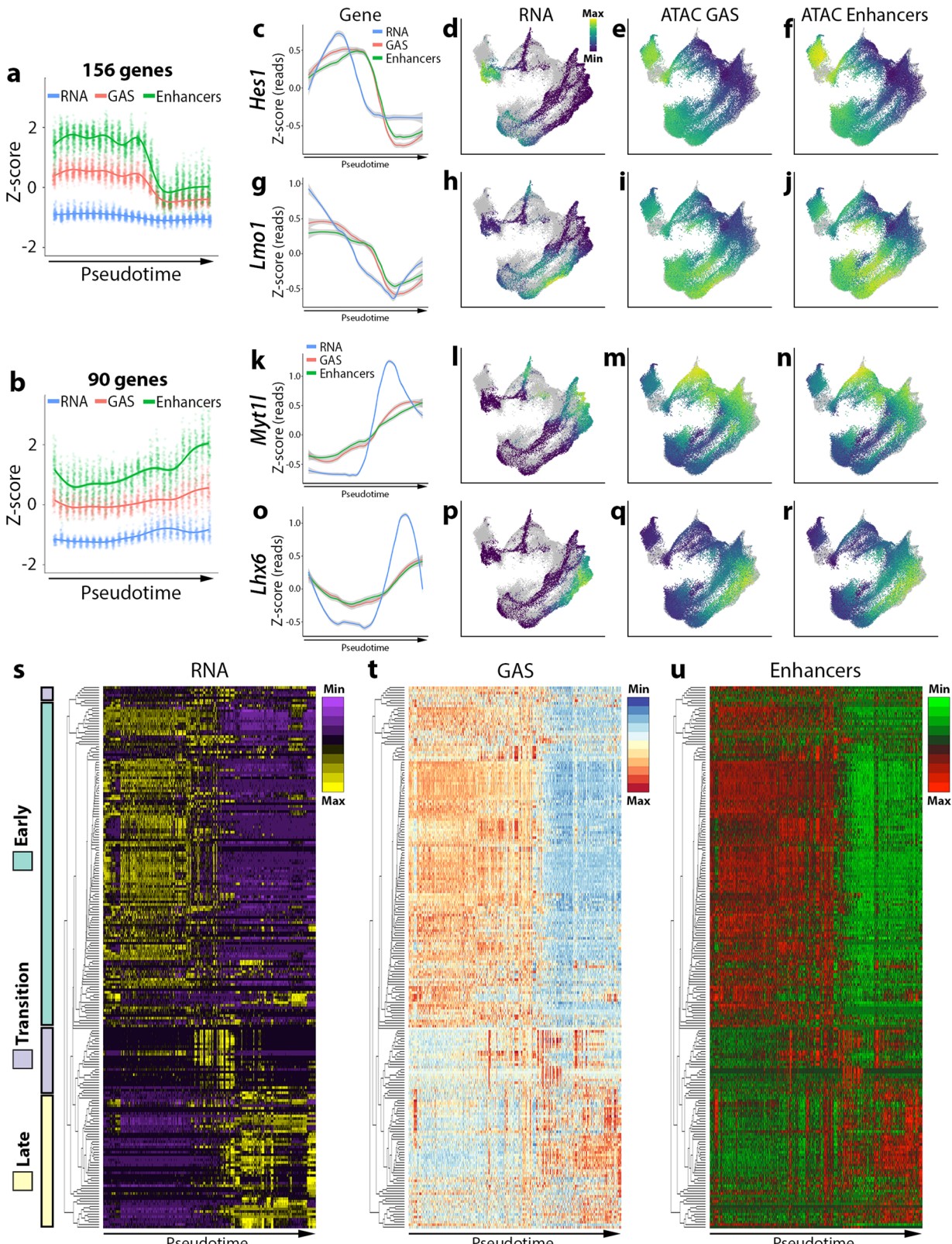

specific to the CGE and MGE, where *Nr2f1* is expressed[36] (Fig. 7d, e). This locus also contains a stronger K27ac signal in the CGE and MGE (Fig. 7f), providing additional evidence for the formation of region-specific promoter–enhancer interactions. Similarly, we observe cortex-enriched interactions of the *Lhx2* promoter with two putative enhancers within *Dennd1a* introns,

with both loci displaying stronger H3K27ac signals in the cortex compared to other regions (Fig. 7g–i).

Thus, the combination of single-cell accessibility and transcriptomes with histone modifications and higher-order chromatin interactions represents a comprehensive epigenomic "ground truth" of distinct neurogenic regions of the embryonic

**Fig. 5 Transcription, gene accessibility, and active enhancer utilization are highly correlated and dynamic during neuron lineage commitment in the mouse forebrain. a, b** Line charts of "early expressed" (**a**) and "late expressed" (**b**) DEG clusters detected by degPatterns using embryonic integrated snATAC-Seq/scRNA-Seq data. Y-axis is Z-score for RNA, GAS, or enhancers counts per gene. The X-axis is binned pseudotime periods. RNA, GAS, or enhancers for individual genes in these clusters are plotted (156 "early" and 90 "late" genes). **c–r** Line chart and UMAP visualizations of RNA, GAS, and enhancer read counts for representative "early" genes *Hes1* and *Lmo1* (**c–j**) and "late" genes *Myt1l* and *Lhx6* (**k–r**). For line charts, Y-axis is Z-score for RNA, GAS, or enhancers counts per gene. The X-axis is binned pseudotime periods. Gray dots in the background of UMAP plots represent cells/nuclei from other assays. In panels **c**, **g**, **k**, **o**, the shaded gray region represents a 95% confidence interval. **s–u** Heatmaps of RNA (**s**), GAS (**t**), and enhancers (**u**) counts for 210 variable genes. Heatmap columns were ordered by hierarchical clustering of 210 variable genes with correlation distance and average linkage. Rows were ordered by pseudotime (assigned by Moncole3). Color bars in **s** indicate genes grouped together based on "early", "transition" and "late" profiles (as assigned by degPatterns and manually refined).

mouse brain that give rise to specific neuronal subtypes. These data are publicly available and searchable as a UCSC Genome Browser track hub (https://www.nichd.nih.gov/research/atNICHD/Investigators/petros/epigenome-atlas).

## Discussion

We characterized the single-cell chromatin accessibility and transcriptomic profiles, histone modifications, and higher-order chromatin organization in four distinct neurogenic regions of the mouse embryonic forebrain. While recent studies performed single-cell sequencing experiments to characterize chromatin accessibility in the mouse and human forebrain[31,50–52], our dataset represents the most comprehensive analysis of the chromatin landscape in the developing brain to date. With this combinatorial approach, we characterized the variation and dynamic reconfiguration of mRNA, gene accessibility, and active enhancers during neurogenesis and across different neurogenic cell types. We identified numerous candidate enhancers for genes involved in well-characterized neuronal subtypes, many with region-specific direct genomic interactions verified by Hi-C and Capture-C. These data are publicly available in an easily searchable platform on the UCSC genome browser (See Data Availability section; https://www.nichd.nih.gov/research/atNICHD/Investigators/petros/epigenome-atlas). This dataset will be an important resource for the field leading to a greater understanding of the genetic and epigenetic mechanisms regulating initial neuronal fate decisions in the embryonic forebrain.

Gene expression and DA peaks were strongly correlated with H3K4me3 and H3K27me3 peaks at active and repressed promoters, respectively, in specific brain regions. There was also a high correlation between ATAC peaks with Cicero connections to gene promoters and H3K27ac marks at these ATAC peaks (Fig. 6b), indicative of likely active enhancers. However, there were genomic loci where not all modalities were in agreement. For example, the promoter of *Nr2f2* is accessible in all four brain regions despite mRNA and the H3K4me3 active promoter mark being restricted to the CGE (Fig. 6b). Thus, the multimodal approach led to a more complete, accurate picture of gene expression and epigenome state compared to looking at one modality alone.

By performing Hi-C and Capture-C on dissected MGE, LGE, CGE, and cortex, we characterized region-specific chromatin domains and enhancer–promoter interactions in vivo that were not previously identifiable in ENCODE or other studies that do not distinguish between different forebrain regions. For example, the *Nkx2-1* chromatin domain is markedly different between the MGE (where *Nkx2-1* directly contacts the *Mbip* locus) and non-MGE (where *Nkx2-1* directly contacts the *Nkx2-9* and *Pax9* locus) (Fig. 7). Perturbing these types of interactions could reveal important insights into how chromatin organization affects promoter–enhancer interactions and gene function in a region-specific manner. As we only examined ~50 genes with Capture-C, the realm of region-specific interactions between genes and cREs

in the developing forebrain is only beginning to be explored. Previous comparative analyses of chromatin structure have described organ and cell type-specific spatial configurations but have focused mostly on adult tissues[53,54]. Analysis of cell populations representing earlier stages of differentiation trajectories have been mostly restricted to the immune system[55,56], limb differentiation[57] and other organisms[58,59] and have revealed lower variation of chromatin structure between different cell types. In contrast, our data show that spatially adjacent cells representing early neuronal specification processes can present vastly heterogeneous 3D chromatin structures.

We note several additional intriguing observations from our data. First, the chromatin accessibility profiles reveal significant diversity in AP clusters from the GEs, much more so than cortical APs which have greater similarity between each other compared to other clusters (Fig. 2). Much of the reported transcriptional diversity within the GEs has been restricted to postmitotic cells[8,9], so our data suggests that there may be greater transcriptional and chromatin state diversity in GE APs than previously appreciated[60]. Second, there was a lag for chromatin at genes and enhancers to become inaccessible compared to RNA down-regulation, both at the individual gene level (Fig. 5c–r) and global level (Fig. 5s–u). This observation implies that some repressive mechanisms (e.g., repressor TFs, DNA methylation, etc.) likely precede repressive histone modifications and decreased chromatin accessibility at cREs. We observed numerous instances where accessibility of promoters, gene body and/or enhancers precede transcript upregulation (e.g., *Nkx2-1* in Fig. 4g, h and *Myt1l* and *Lhx6* in Fig. 5k–r), which is in agreement with several recent reports[51,61]. Future multiomics studies that can simultaneously capture the epigenome and transcriptome within single cells during development should provide significant insight into this relationship.

Third, the number and score of global Cicero connections near a particular gene (orange arcs from Fig. 3b, c) are only loosely correlated to TSS-cRE connections. In some instances, the number of global Cicero connections at certain genomic loci appear to be tissue-specific (greater co-accessibility in regions flanking *Neurod6* restricted to cortex and *Lhx6* restricted to MGE), while global Cicero connections near other genes appear more similar between tissues (similar co-accessibility for all brain regions flanking *Sp8* and *Nr2f2*). This may suggest a role of Cis-Co-accessibility Networks (CCANs)[37], modules of connection sites that are highly co-accessible with one another in specific brain regions during neuronal fate determination.

Fourth, the population of "mixed" cells that were collected with the LGE and CGE tissue expressed markers for both GABAergic and glutamatergic cells yet formed a distinct cluster from the cortex and GE populations (Fig. 2 and Supplementary Fig. 2). These "mixed" cells likely reside at the PSB as they were not detected in the MGE population. The diversity of cells arising from the lateral/ventral pallium remains poorly characterized, but this region appears to give rise to glutamatergic cells of the insular

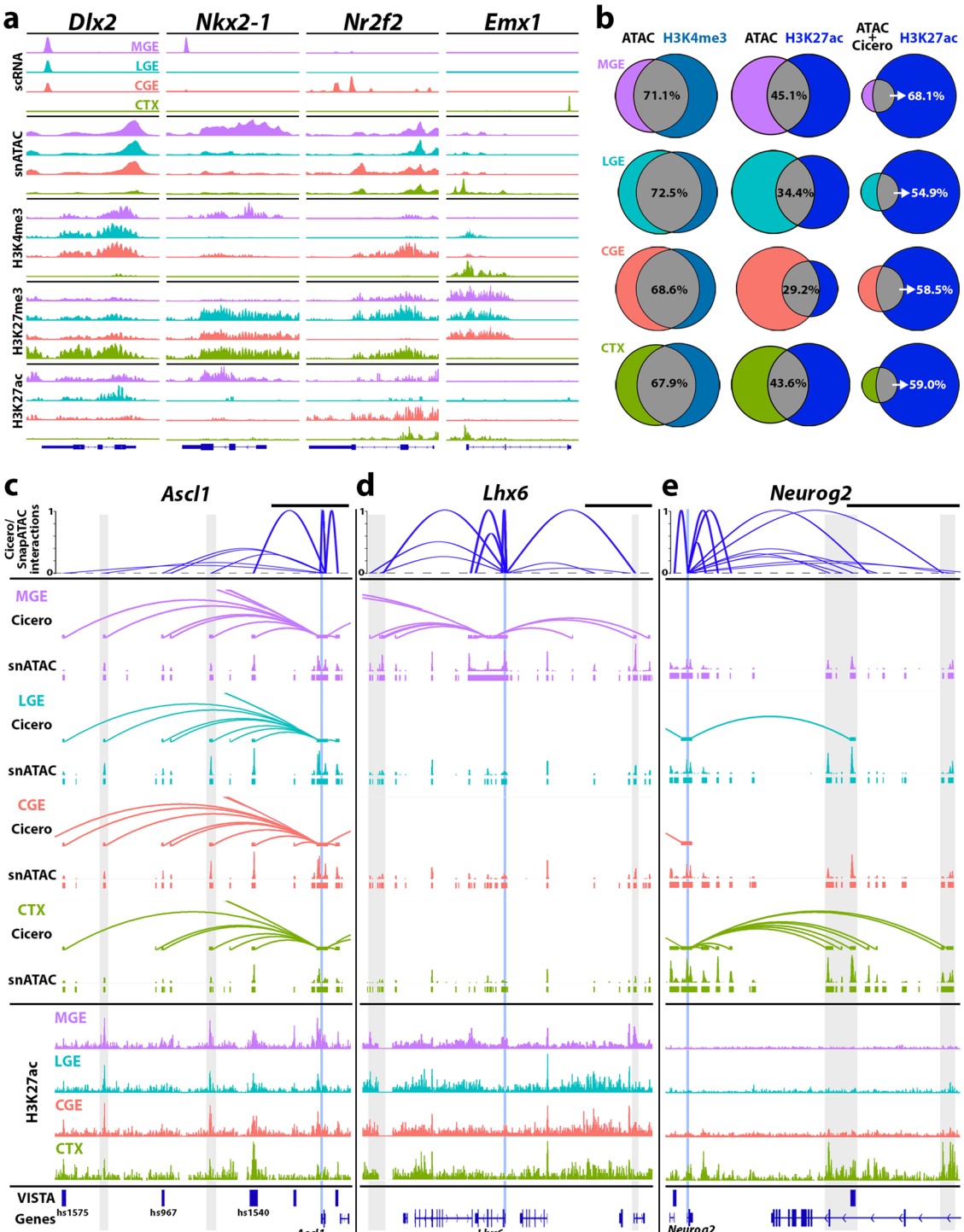

**Fig. 6 Histone modifications within mouse embryonic neural progenitors. a** Tracks of scRNA-Seq, snATAC-Seq, and histone modifications correlated with active promoters (H3K4me3), repressed genes (H3K27me3), and active enhancers (H3K27ac) in MGE, LGE, CGE, and cortex (CTX); pan-GE *Dlx2*, MGE-restricted *Nkx2-1*, CGE-enriched *Nr2f2*, and cortex-restricted *Emx1*. **b** Venn diagrams comparing ATAC peaks at promoter regions vs. H3K4me3 peaks at promoters (left); ATAC peaks outside promoters vs. H3K27ac peaks outside promoters (middle); and ATAC peaks with Cicero connections to a gene promoter vs. H3K27ac peaks outside promoters. Percentages represent % of ATAC peaks overlapping with histone marks per brain region. Circle size represents the relative number of peaks in each group. **c-e** Top, Integration of gene-enhancer predictions using Cicero/SnapATAC interactions as in Supplementary Fig. 6. Arc height of Cicero/SnapATAC interactions track indicates relative interaction scores between gene TSS and predicted cis-regulatory elements. Middle, tissue-specific Cicero connections with snATAC peaks. Cicero connections were filtered to retain scores >0.25, and connections where one anchor intersects a gene TSS while the second anchor does not intersect promoter regions of any genes. Bottom, H3K27ac tracks with VISTA hits for *Ascl1* (**c**), *Lhx6* (**d**), and *Neurog2* (**e**). Vertical light blue line denotes TSS for each gene, gray-shaded rectangles indicate loci of interest related to TSS. VISTA hits near genes are depicted with dark blue bars. Black scale bars above Cicero/SnapATAC tracks = 50 kb.

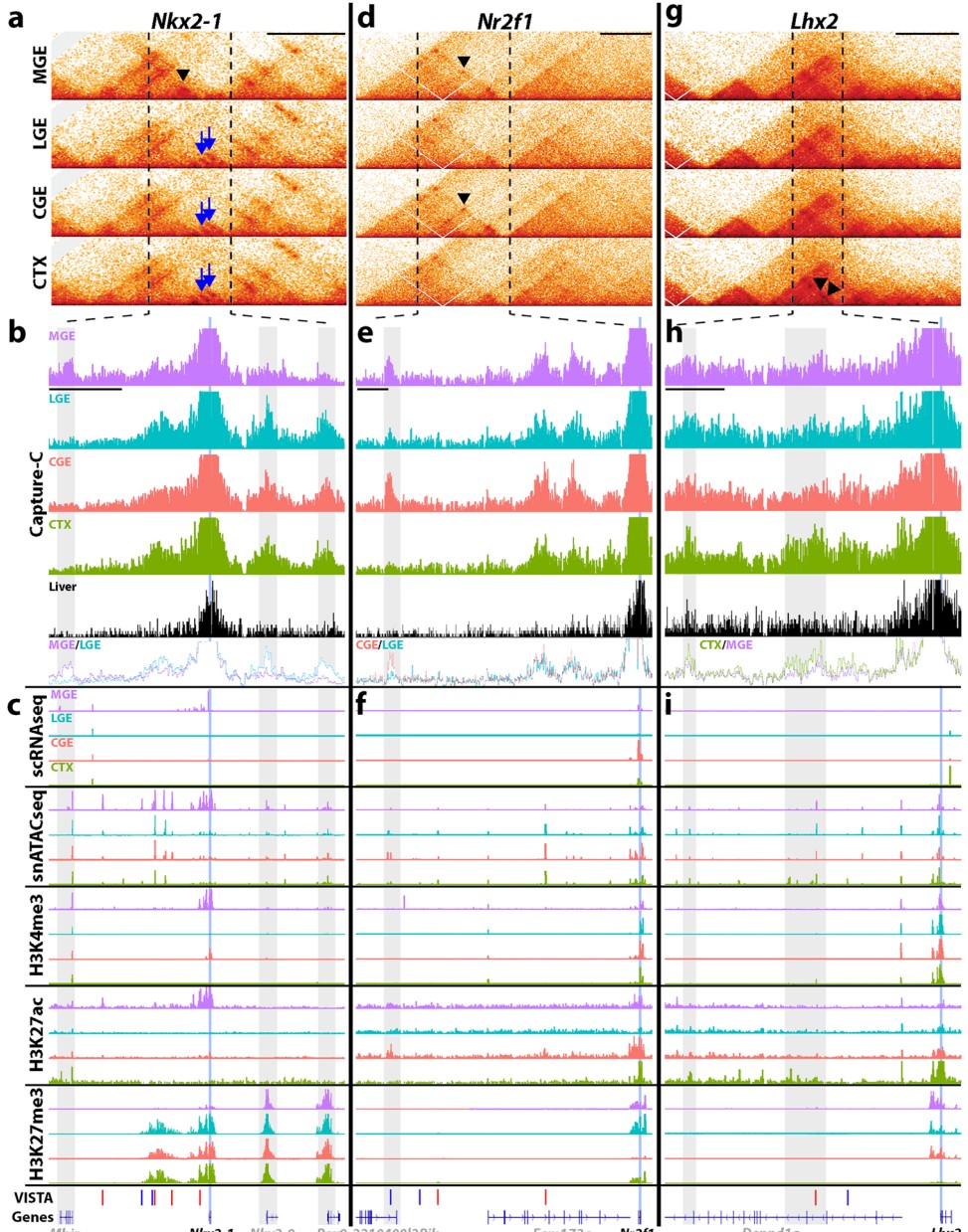

**Fig. 7 Higher-order chromatin structure within mouse embryonic neural progenitors. a** Hi-C 1D interaction frequency heatmaps for MGE, LGE, CGE, and CTX at MGE-restricted *Nkx2-1*. Black triangle denotes an MGE-specific interaction, blue arrows denote non-MGE-specific interactions. **b** Capture-C pileup at *Nkx2-1* locus extending to vertical dashed lines in **a**. Y-axis represents reads from loci interacting with *Nkx2-1* promoter bait. Blue line indicates gene TSS, gray bars indicate potential regulatory elements directly interacting with the *Nkx2-1* promoter. **c** Signal tracks for single-cell assays (scRNA-Seq and snATAC-Seq) and histone modifications correlated with active promoters (H3K4me3), active enhancers (H3K27ac), and repressed genes (H3K27me3) in each tissue. VISTA track includes tested enhancers. **d–f** Hi-C heatmaps (**d**), Capture-C interactions (**e**), and chromatin landscape (**f**) of CGE-enriched *Nr2f1*. Black triangles in **d** denote MGE and CGE-enriched interactions. **g–i** Hi-C heatmaps (**g**), Capture-C interactions (**h**), and chromatin landscape (**i**) of cortex-restricted *Lhx2*. Black triangles in **g** denote CTX-enriched interactions. Black scale bars above Hi-C plots in **a**, **d**, **g** = 500 kb, black scale bars below MGE Capture-C tracks in **b**, **e**, **h** = 100 kb.

cortex, piriform cortex, claustrum, and pallial amygdala[26]. A recent scRNA-seq study may shed light on the heterogeneity of this PSB region[27], but further work is needed to better characterize this cell population.

In sum, the single-cell chromatin accessibility and transcript profiles, histone modifications, and higher-order chromatin structure define the epigenetic "ground truth" of distinct forebrain regions during initial neuronal fate decisions. This resource will aid our understanding of normal development and

neurological disease as many disease-associated genes are enriched in neural progenitors and immature neurons[19,20], and many disease-associated SNPs are located in non-coding enhancer regions[21–23].

## Methods

**Animals**. All experimental procedures were conducted in accordance with the National Institutes of Health guidelines and were approved by the *Eunice Kennedy Shriver* NICHD Animal Care and Use Committee (protocol #20-047). The

following mouse lines were used in this study: C57BL/6 J (JAX# 000664). For timed matings, noon on the day a vaginal plug was observed was denoted E0.5. For each experimental modality, brain regions from multiple embryos (≥4) were pooled together prior to single-cell dissociations. Both male and female embryonic mice were used without bias for all experiments. Housing conditions: 12/12 h light/dark cycle, humidity between 30–50%, temperature 72 °C.

**Tissue dissection**. To recover embryonic tissue, dams were anesthetized with isoflurane and then euthanized by cervical dislocation. Embryos were removed from the uterus and kept in chilled artificial cerebrospinal fluid (ACSF, in mM: 87 NaCl, 26 NaHCO$_3$, 2.5 KCl, 1.25 NaH$_2$PO$_4$, 0.5 CaCl$_2$, 7 MgCl$_2$, 10 glucose, 75 sucrose, saturated with 95% O$_2$, 5% CO$_2$, pH 7.4). Brains were extracted from E12.5 and E14.5 embryos, hemisected, microdissected to obtain the MGE, LGE, CGE, and somatosensory cortex, and kept in ACSF.

**Nuclei extraction for single nuclei ATAC-seq, CUT&Tag, and CUT&RUN**. Nuclei isolation followed the 10X Genomics ATAC nuclei isolation protocol with several modifications. All steps were performed on ice. For each brain region, tissue was transferred to a Dounce homogenizer containing 1 mL ATAC lysis buffer (10 mM Tris-HCL, pH 7.4, 10 mM NaCl, 3 mM MgCl2, 0.1% Tween-20, 0.1% IGEPAL, 2% BSA). Samples were dounced ten strokes pestle A and ten strokes pestle B. Lysate was strained through a 40-μm filter pre-wetted with ATAC wash buffer (10 mM Tris-HCL, pH 7.4, 10 mM NaCl, 3 mM MgCl2, 0.1% Tween-20, 2% BSA) and neutralized with 2 mL wash buffer.

For snATAC-Seq, lysates were centrifuged 500 × g for 5 min at 4 °C, the supernatant was removed, the nuclei pellet washed once with 1 mL wash buffer, and centrifuged 500×g for 5 min. A diluted aliquot of nuclei solution was mixed with Trypan Blue (1:1) and counted on a hemocytometer. For all samples, we prepared 3000 nuclei/μL samples for snATAC-seq reactions, with 5 μL (~15,000 nuclei) used for each snATAC reaction.

For CUT&Tag/CUT&RUN, tissue was homogenized as described above. Nuclei suspensions were centrifuged 500 × g for 5 min at 4 °C, the supernatant removed, and washed once with 1 mL wash buffer, centrifuged 500 × g for 5 min, and washed a final time with 1 mL 1X CUT&Tag wash buffer (from CUT&Tag IT Assay Kit, Active Motif, #53610) or 1 mL CUT&RUN 1X wash buffer (1 mL HEPES pH 7.5, 1.5 m 5 M NaCl, 12.5 uL 2 M spermidine and 47.5 mL dH$_2$O with 1 Roche Complete Protease Inhibitor EDTA-Free tablet). A diluted aliquot of nuclei solution was mixed with Trypan Blue (1:1) and counted on a hemocytometer. Equal numbers of MGE, CGE, LGE, and cortex nuclei were pooled and diluted to a final concentration of 1000 nuclei/μL in wash buffer, with 100 μL (~100,000 nuclei) used for each CUT&Tag/CUT&RUN reaction.

**Cell dissociation for single-cell RNA-Seq, Hi-C, and Capture-C**. Embryonic tissue was dissected as described above. To collect whole cells, embryonic MGE, LGE, CGE, and cortex tissue was incubated in 1 mg/mL Pronase (Roche #10165921001) in ACSF for 20 min at RT. Pronase solution was removed and 2 ml of reconstitution solution (1% fetal bovine serum + DNAse (1:10,000, Roche #47167280001)) in oxygenated ACSF was added. For Hi-C and Capture-C preparations, DNAse was not included in the reconstitution solution. Cells were triturated sequentially with fire-polished large, medium, and small-bore Pasteur pipettes to mechanically dissociate tissue.

For scRNA-Seq, DRAQ5 (20 µM) and DAPI (1:10,000) was added to the single-cell suspension, which was passed through a pre-wet 30-µm filter and then processed on an SH800 cell sorter to purify the sample. DRAQ5 + /DAPI—live cells were collected in low-bind 1.5 mL tubes containing 100 µL ACSF. Cell solutions were centrifuged at 300 × g for 5 min at 4 °C in a swinging bucket centrifuge and then counted on a hemocytometer. About 15,000 cells (or the highest amount recovered after sorting) was used for 10X Genomics scRNA-Seq experiments.

**snATAC library preparation and sequencing**. snATAC reaction was carried out following 10X Genomics ATAC User Guide (revision C), libraries were prepared following 10X Genomics and Illumina guidelines, and sequenced on an Illumina HiSeq2500. Sequencing metrics were as follows: CGE: Replicate 1: Read pairs: 469,280,227; Estimated number of cells: 4013; Median fragments per cell: 13,003; Fraction of fragments in peaks: 73.3%; Fraction of transposition events in peaks: 58.4%. Replicate 2: Read pairs: 115,917,541; Estimated number of cells: 4530; Median fragments per cell: 11,082; Fraction of fragments in peaks: 80.3%, Fraction of transposition events in peaks: 72.1%. MGE: Replicate 1: Read pairs: 429,523,963; Estimated number of cells: 6845; Median fragments per cell: 12,807; Fraction of fragments in peaks: 72.0%, Fraction of transposition events in peaks: 54.2%. Replicate 2: Read pairs: 107,989,853; Estimated number of cells: 3465; Median fragments per cell: 12,278; Fraction of fragments in peaks: 77.4%, Fraction of transposition events in peaks: 67.3%. LGE: Replicate 1: Read pairs: 491,904,518; Estimated number of cells: 6577; Median fragments per cell: 13,426; Fraction of fragments in peaks: 67.7%, Fraction of transposition events in peaks: 47.4%. Replicate 2: Read pairs: 115,545,951; Estimated number of cells: 4769; Median fragments per cell: 10,660; Fraction of fragments in peaks: 77.7%, Fraction of transposition events in peaks: 65.8%. Cortex (E12.5): Read pairs: 112,120,408;

Estimated number of cells: 4946; Median fragments per cell: 11,257; Fraction of fragments in peaks: 78.0%, Fraction of transposition events in peaks: 68.1%. Cortex (E14.5): Read pairs: 433,510,039; Estimated number of cells: 4108; Median fragments per cell: 16,810; Fraction of fragments in peaks: 74.2%, Fraction of transposition events in peaks: 63.6%.

**scRNA library preparation and sequencing**. cDNA libraries were prepared using 10X Genomics 3′ RNA v3 chemistry. Library preparation was carried out following the 10X Genomics RNA User Guide (rev C) and sequenced following 10X Genomics and Illumina guidelines. Samples were sequenced to the following depths: CGE: Reads: 154,348,231; Estimated number of cells: 4522; Median reads per cell: 34,133; Median genes per cell: 2805. MGE: Reads: 192,833,398; Estimated number of cells: 6331; Median reads per cell: 30,459; Median genes per cell: 2877. LGE: Reads: 180,580,907; Estimated number of cells: 6843; Median reads per cell: 26,389; Median genes per cell: 2726. Cortex (E12.5): Reads: 176,542,467; Estimated number of cells: 7453; Median reads per cell: 23,687; Median genes per cell: 2318.

**CUT&Tag library preparation and sequencing**. Embryonic tissue was homogenized as described above before proceeding with CUT&Tag. For each CUT&Tag replicate, 100,000 nuclei were resuspended in 1.5 mL 1X Wash Buffer and then processed with the Active Motif CUT&Tag IT Assay Kit (#53610) following manufacturer's instructions. Primary antibodies used: rabbit anti-H3K4me3 (Active Motif 39159, 1:50), rabbit anti-H3K27me3 (Cell Signaling 9733 T, 1:50). Secondary antibody used was guinea pig anti-rabbit (Active Motif 105465 from CUT&Tag IT Assay Kit, 1:100). Following library amplification, DNA quantity was determined with a Thermo Qubit and library quality characterized with an Agilent Tapestation. Libraries were balanced for DNA content and pooled before performing a final SPRIselect bead 1x left size selection and paired-end sequenced (50 × 50 bp) on an Illumina NovaSeq. Samples were sequenced to a following depths per library: CGE: H3K27me3 Replicate 1: 13,604,105; H3K27me3 Replicate 2: 47,136,470; H3K4me3 Replicate 1: 50,541,513; H3K4me3 Replicate 2: 25,035,394; CTX: H3K27me3 Replicate 1: 13,640,572; H3K27me3 Replicate 2: 36,097,462; H3K4me3 Replicate 1: 18,35,5546; H3K4me3 Replicate 2: 30,114,008; LGE: H3K27me3 Replicate 1: 13,539,124; H3K27me3 Replicate 2: 30,498,855; H3K4me3 Replicate 1: 14,760,336; H3K4me3 Replicate 2: 17,211,681; MGE: H3K27me3 Replicate 1: 17,214,723; H3K27me3 Replicate 2: 16,336,430; H3K4me3 Replicate 1: 5,518,769; H3K4me3 Replicate 2: 14,720,921.

**CUT&RUN library preparation and sequencing**. Embryonic tissue was homogenized as described above. Single nuclei suspensions were centrifuged 500×g for 5 min at 4 °C, the supernatant removed, and washed with 1 mL 1X Wash Buffer (1 mL HEPES pH 7.5, 1.5 m 5 M NaCl, 12.5 uL 2 M spermidine and 47.5 mL dH$_2$O with 1 Roche Complete Protease Inhibitor EDTA-Free tablet). For each CUT&RUN replicate, 100,000 nuclei were resuspended in 1.5 mL 1X Wash buffer. BioMag$^®$ Plus Concanavalin A beads (Bangs Laboratories) were washed in in Binding Buffer (20 mM HEPES pH 7.5, 10 mM KCl, 1 mM CaCl2, 1 mM MnCl2). Nuclei were resuspended in Wash Buffer, mixed with a slurry of the Concavalin A coated magnetic beads, and rotated for 10 min at room temperature. 10 µl of Concavalin A bead slurry was used per 100,000 cells. The beads were resuspended in Wash Buffer containing 2 mM EDTA, 0.1% bovine serum albumin, 0.05% Digitonin, and 1:50 dilution of primary antibody (rabbit anti-H3K27ac, Abcam ab4729), which was then incubated on a nutating platform for 2 h at room temperature. The beads were then washed twice in Digitonin Buffer (20 mM HEPES pH 7.5, 150 mM NaCl, 0.5 mM Spermidine, 1x Roche Complete Protease Inhibitor no EDTA, 0.05% Digitonin, and 0.1% bovine serum albumin). Then they were incubated with pA-MN (600 µg/ml, 1:200, either homemade or a gift from S. Henikoff) in Digitonin Buffer for 1 h at 4 °C. Following this incubation, beads were washed twice with Digitonin Buffer and finally resuspended in 150 µl of Digitonin Buffer, and equilibrated to 0 °C before adding CaCl2 (2 mM). The beads were then incubated for 1 h at 0 °C. After this hour, 150 µl of 2X Stop Buffer (200 mM NaCl, 20 mM EDTA, 4 mM EGTA, 50 µg/ml RNase A, 40 µg/ml glycogen), was added. Beads were incubated for 30 min at 37 °C and then pelleted at 16,000 × g for 5 min at 4 °C. Supernatant was transferred, mixed with 3 µL 10% SDS and 1.8U Proteinase K (NEB), and incubated for 1 h at 50 °C, shaking at 900 rpm. About 300 µl of 25:24:1 Phenol/Chloroform/Isoamyl Alcohol was added, solutions were vortexed, and transferred to Maxtrack phase-lock tubes (Qiagen). The samples in the phase-lock tubes were centrifuged at 16,000 × g for 3 min at room temperature. About 300 µl of Chloroform was added, and solutions were mixed by inversion and centrifuged at 16,000 × g for 3 min at room temperature. Aqueous layers were transferred to new tubes and DNA was isolated through Ethanol precipitation. These samples were resuspended in 10 mM Tris-HCl pH 8.0 (Thermo Fisher). CUT&RUN libraries were prepared following the SMARTer ThruPlex TAKARA Library Prep kit with small modifications. Double-stranded DNA (10 µl), Template Preparation D Buffer (2 µl), and Template Preparation D Enzyme (1 µl) were combined and added to each sample. End Repair and A-tailing was performed in a Thermocycler with a heated lid (22 °C, 25 min; 55 °C, 20 min). To each sample, library Synthesis D Buffer (1 µl) and Library Synthesis D Enzyme (1 µl) and library synthesis was performed (22 °C, 40 min). Library Amplification D Buffer (25 µl), Library Amplification D Enzyme (1 µl), Nuclease-free water (4 µl), and a unique

Illumina-compatible indexed primer (5 µl) were added. Library amplification was performed using the following conditions: 72 °C for 3 min; 85 °C for 2 min; 98 °C for 2 min (denaturation); four cycles of 98 °C for 20 s, 67 °C for 20 s, 72 °C for 10 s (addition of indexes); 14 cycles of 98 °C for 20 s, 72 °C for 10 s (library amplification). Post-PCR clean-up involved SPRIselect bead 0.6X left/1x right double size selection then washed twice gently in 80% ethanol and eluted in 10–12 µl 10 mM Tris pH 8.0. 1:50 dilution of the primary antibody was used. Following library amplification, DNA quantity was determined with a Thermo Qubit and library quality was characterized with an Agilent Tapestation. Libraries were balanced for DNA content and pooled before performing a final SPRIselect bead 1x left size selection and paired-end sequenced (50 × 50 bp) on an Illumina NovaSeq. CGE: H3k27ac Replicate 1: 39,950,806; H3k27ac Replicate 2: 68,704,235; CTX: H3k27ac Replicate 1: 45,279,956; H3k27ac Replicate 2: 73,920,447; LGE: H3k27ac Replicate 1: 46,933,225; H3k27ac Replicate 2: 55,438,848; MGE: H3k27ac Replicate 1: 55,871,849; H3k27ac Replicate 2: 77,034,587.

**Hi-C and Capture-C library preparation and sequencing.** Hi-C and Capture-C were performed and analyzed as described previously[62]. Embryonic tissue was dissected and cells were dissociated as described above. After dissociation, 1 million cells per region were fixed with 1% formaldehyde (Thermo: 28908) made in 1 ml HBSS media. Fixation was carried out at room temperature on a nutator, for 10 min, protected from light. To stop fixation, glycine was added at a final concentration of 0.13 M and samples were incubated for 5 min at room temperature followed by 15 min on ice. Fixed cells were then washed once with ice-cold PBS. After spinning cells at 2500×g 4 °C for 5 min, the pellet was flash-frozen in liquid nitrogen and stored at −80 °C. To perform Hi-C and Capture-C, pellets were first thawed on ice and then incubated with 1 ml lysis buffer (10 mM Tris-HCL pH 8, 10 mM NaCl, 0.2% Igepal CA-630, Roche Complete EDTA-free Sigma #11836170001). After lysis, cells were dounced and washed with cold PBS. Nuclei extracted with this method were then collected by centrifugation and subjected to DpnII digest in 50 µl 0.5% SDS and incubated at 62 °C for 10 min after which 150 µl of 1.5% Triton-X was added and cells incubated for 15 min at 37 °C while shaking at 900 rpm. Twenty-five microliters of 10X DpnII restriction buffer (NEB) was then added, and cells were incubated for 15 min while shaking. After that, 200 U of DpnII (NEB R0543M) were added and incubated for 2 h, then 200 U more and incubated overnight. The next morning 200 U more were added and incubated for 3 h (total of 600 U of DpnII). DpnII was inactivated at 62 °C for 20 min. For Hi-C, biotin fill-in was done by incubating cells with a mixture of 4.5 µl dCTP dTTP and dGTP at 3.3 mM, 8 µl Klenow polymerase (NEB M0210L), and 37.5 µl Biotin-14-dATP (Thermo 19524016) for 4 h at RT while shaking at 900 rpm for 10 s every 5 min. Ligation was done overnight at 16 °C also rotating at 900 rpm for 10 s every 5 min by adding 120 µl of 10X ligation buffer (NEB), 664 µl water, 100 µl 10% Triton-X, 6 µl BSA 20 mg/ml, and 2 µl T4 ligase (NEB cat #M0202M). For Capture-C, the biotin fill-in step was skipped and 50 µl more of water was added to the ligation mix. Crosslink removal was done overnight with 50 µl of proteinase K in 300 µl of the following buffer (10 mM Tris-HCl pH 8.0, 0.5 M NaCl, 1% SDS) while shaking at 1400 rpm at 65 °C. Following Sodium Acetate and 100% Ethanol −80°C precipitation, DNA was resuspended in 50 µl 10 mM Tris-HCL for Hi-C or 130 µl for Capture-C. Sonication for Hi-C was done using Covaris onetube-10 AFA strips using the following parameters for a 300 bp fragment size (Duration: 10 s, repeat for 12 times, total time 120 s, peak power-20 W, duty factor 40%, CPB-50). Sonication for Capture-C was done using Covaris AFA microtubes 130 with the following settings for a fragment size of 200 bp fragments (Duration: 225 s, peak power-75 W, duty factor 25%, Cycles per Burst-1000). Sonications were performed in a Covaris ME220 sonicator. Sonicated material was then size selected using SPRI beads with the following ratios: 0.55X and 1X for Capture-C and 0.55X and 0.75X for Hi-C. Hi-C material was then bound, washed, and recovered to 150 µl Streptavidin C1 beads (Thermo 65002) per sample following manufacturers' recommendations. Bead-bound DNA was resuspended in 50 µl 10 mM Tris-HCl. Library preparation was done using the Kapa Hyper Prep KK8502 kit. Ten microliters of End-repair buffer and enzyme mix were added to resuspended beads and incubated for 30 min at RT and then 30 min at 65 °C. One microliter of 15 mM annealed-Illumina adapters, containing a universal p5 and an indexed p7 oligo, were then incubated with a mixture containing 40 µl of ligase and ligation buffer at RT for 60 min. Libraries were then amplified using four reactions per sample for a total of 200 µl and ten cycles, as recommended by the manufacturer. For Capture-C, following sonication and size selection, 1 µg of template material was resuspended in 50 µl of 10 mM Tris and used for library prep with 10 µl of End-Repair reaction. Five microliters of 15 mM annealed -Illumina adapters were ligated to the Capture-C material. Using a total volume of 100 µl, the library was amplified by PCR using six cycles. For capture, 1 µg of Capture-C library per sample was mixed with mouse COT1 DNA and universal as well as index-specific blocking oligos from SeqCap EZ HE-oligo (Roche). About 4.5 µl pool of biotinylated probes (xGen Lockdown Probe Pools from IDT), with each probe at 0.4 fmol/µl targeting the promoters of our loci of interest were added to this mixture and incubated for 3 days at 47 °C. Following binding to Streptavidin C1 beads, the material was washed as recommended by the SeqCap EZ Hybridization and Wash Kits. Following washes material was amplified by PCR using Kapa polymerase and 14 cycles. Material from different samples was then combined and 1 µg of pooled libraries was recaptured in a single reaction and amplified with eight cycles. Probes

for Capture-C were designed using Capsequm (capsequm.molbiol.ox.ac.uk/cgi-bin/CapSequm.cgi) by selecting regions adjacent to the gene promoters of interest. The list of probes used can be found in Supplementary Data 7.

**scRNA-seq analysis.** *Cellranger*: The Cellranger (v3.0.0) pipeline was used to process single cell/nuclei RNA-Seq libraries, all steps used default parameters unless otherwise noted. Cellranger mkfastq converted BCL files generated from sequencing to demultiplexed FASTQ files. Reads were aligned to 10X Genomics's pre-built mouse (GRCm38/mm10) reference genome with Cellranger count. For single-cell RNA-Seq libraries, 10X Genomics's pre-built mRNA model of GRCm38/mm10 annotation (refdata-cellranger-mm10-3.0.0) was used to map reads to genes, while single nuclei RNA-Seq libraries used a corresponding pre-mRNA model constructed per 10X Genomics guidelines. Aligned reads were de-duplicated, filtered for valid cell barcodes, and used to construct gene-by-barcode matrices.

*Seurat*: Filtered gene-by-barcode matrices were used as input to Seurat (v3.0.0, https://satijalab.org/seurat/) in R (v.4.0.0, https://cran.r-project.org). For each cell barcode, summary statistics were calculated for the metadata columns n_Features, n_genes, and percent reads mapping to mitochondrial genes (if present). Outliers within the metadata columns were detected using Tukey's fence method for far-out outliers ([Q1 − k(Q3 − Q1), Q3 + k(Q3 − Q1)], where k = 3) which is resistant to extreme values (https://datatest.readthedocs.io/en/stable/how-to/outliers.html) and barcodes with any metadata column that contained outliers were removed. The remaining barcodes were processed using the SCTransform workflow in Seurat.

**snATAC-seq analysis.** *Cellranger-atac*: The Cellranger-atac (v1.2.0) pipeline was used to process single nuclei ATAC-Seq data, all steps used default parameters unless otherwise noted. Cellranger-atac mkfastq converted BCL files generated from sequencing to demultiplexed FASTQ files. Reads were aligned to 10X Genomics's pre-built mouse (GRCm38/mm10) reference genome and genomic annotation (refdata-cellranger-atac-mm10-1.2.0) by Cellranger-atac count. Libraries were aggregated and downsampled to equal numbers of median fragments per nuclei by Cellranger-atac aggr. Aligned reads were de-duplicated, filtered for valid cell barcodes, and constructed peak-by-barcode and TF-by-barcode matrices and fragments file.

*Signac*: Filtered peak-by-barcode matrix and fragments file were analyzed using Signac (v1.0.0) (https://satijalab.org/signac/index.html), all steps used default parameters unless otherwise noted. For each cell, summary statistics were calculated for the metadata columns n_Features, n_genes, and percent reads mapping to mitochondrial genes (if present). Outliers within the metadata columns were detected using Tukey's fence method for far-out outliers, as previously described, and barcodes with any metadata column that contained outliers were removed.

Following pre-processing, peak-by-barcode matrices were processed as instructed by Signac documentation: highly variable peaks were detected by Signac::FindTopFeatures() and peak-by-barcode matrices normalized by Term Frequency–Inverse Document Frequency (TF-IDF) method by Signac::RunTFIDF().

To address batch specific variation, the integration framework from the Seurat package was used on normalized peak-by-barcode matrices with the functions Seurat::FindIntegrationAnchors(), which detects features common to all batches, and Seurat::IntegrateEmbeddings(), which computes a weights matrix using the low dimensional cell embeddings (LSI coordinates), prior to merging counts matrices across batches. All downstream Seurat/Signac functions included "latent.vars" arguments to further regress out confounding variables.

Following integration steps, dimensional reduction with Singular Value Decomposition (SVD) and Uniform Manifold Approximation and Projection (UMAP), and cluster detection using smart local moving (SLM) algorithm were performed. RNA cell type predictions from scRNA-Seq were transferred to snATAC-Seq nuclei using the label transfer method outlined by Signac. Following the standard Signac workflow, promoter accessibility (PA) for each gene was calculated by summing all reads mapping to a gene body plus promoter (2 Kb upstream from gene TSSs). Differential testing was performed for peak counts and PA by Seurat::FindAllMarkers(min.pct=0.2, test.use = "LR", latent.vars = "nCount_peaks") and Seurat::FindAllMarkers(), respectively. Motif analysis was performed using Signac by Seurat::FindMarkers(only.pos=TRUE, min.pct=0.2, test.use = "LR", latent.vars = "nCount_peaks") followed by Signac::FindMotifs() and using the JASPAR (https://jaspar.genereg.net) CORE vertebrates collection as a reference database. Per cell motif deviation scores were computed using chromVAR (v1.12.0) and the UCSC mm10 genome sequences (BSgenome.Mmusculus.UCSC.mm10, v1.4.0), both from Bioconductor (v3.12), with the Signac::RunChromVAR() wrapper.

*Cicero*: Following outlier detection and dimensional reduction, the filtered peak-by-barcode counts matrix was transferred from the Seurat object to a Monocle3 (https://cole-trapnell-lab.github.io/monocle3/) cell_data_set (CDS) object. The counts object was processed with Cicero (v1.9.1) (https://cole-trapnell-lab.github.io/cicero-release/) following user documentation with defaults unless otherwise specified. Prior to calculating Cicero co-accessibility scores, a Cicero CDS object was created using the newly created Monocle3 cds object and UMAP coordinates

from the Seurat object (i.e., cicero::make_cicero_cds(cds.object, reduced_coordinates=seurat.object@reductions$umap@cell.embeddings)).

*SnapATAC*: snATAC-Seq reads were processed with SnapATAC (v1.0.0) independently of Seurat/Signac/Cicero workflows following user documentation (https://github.com/r3fang/SnapATAC) with default parameters unless otherwise specified. FASTQ reads were converted into snap files using Snaptools (v1.4.1) and Python 2.7 and ATAC accessible regions were binned into 5000 kB windows and counted to create cell-by-bin matrices. Snap files were analyzed by SnapATAC. Valid cell barcodes with log10(UMI) between 3 to 6, and promoter ratio between 0.05 to 0.6 were retained for further analysis. All other barcodes were discarded. cell-by-bin matrices were merged, binarized, filtered against ENCODE blacklist regions (http://mitra.stanford.edu/kundaje/akundaje/release/blacklists/), and barcodes with bin counts less than 1000 and greater than the 95th percentile were removed. Dimensional reduction was performed using Nyström landmark diffusion maps algorithm, batch reduction was performed using Harmony (https://github.com/immunogenomics/harmony) by SnapATAC::runHarmony(eigs.dim = 1:40), a K Nearest Neighbor (KNN) graph constructed by SnapATAC::runKNN(eigs.dims = 1:dims_use, k = 15) and the Louvain algorithm used to detect clusters. Gene activity scores were calculated by counting snATAC-Seq reads mapping to known gene bodies. scRNA-Seq/snRNA-Seq based cell type predictions were transferred to corresponding snATAC-Seq barcodes by SnapATAC::snapToSeurat(eigs.dims = 1:40,norm = TRUE,scale = TRUE), Seurat::FindTransferAnchors() and Seurat::TransferData() as described in SnapATAC documentation. Cluster specific peaks were detected using MACS2 by SnapATAC::runMACS(macs.options = "--nomodel --shift 100 --ext 200 --qval 5e-2 -B --SPMR"). Gene-enhancer pairs were predicted by SnapATAC::predictGenePeakPair() for every TSS that had a nonzero peak count across all barcodes. The gene-enhancer pairs list was filtered by removing pairs where FDR >0.05.

*Enhancer Prediction*: The greatest SnapATAC −log10(adjusted *p* value) value was multiplied by 2 as an upper limit, and all −log10(adjusted *p* values) were percentile ranked per gene. Similarly, Cicero co-accessibility scores were percentile ranked for each gene. Peaks detected by SnapATAC or Cicero that did not overlap gene TSSs were extended 500 bp upstream and downstream before merging overlapping peak coordinates using the GenomicRanges (v1.42.0) Bioconductor package with GenomicRanges::reduce(). To generate a list of all presumptive enhancers, merged SnapATAC peaks and merged Cicero peaks were concatenated. To generate a more stringent list of enhancers for snATAC-Seq/scRNA-Seq integration, the intersection SnapATAC and Cicero peaks was taken by GenomicRanges::intersect(). To detect H3K27ac+ presumptive enhancers, concatenated peaks were further filtered by retaining only peaks that intersected forebrain H3K27ac ChIP-Seq peaks for age E12.5 (UCSC Database: mm10, Primary Table: encode3Ren_forebrain_H3K27ac_E12, Big Bed File: /gbdb/mm10/encode3/histones/ENCFF957YEE.bigBed) or E14.5 (UCSC Database: mm10, Primary Table: encode3Ren_forebrain_H3K27ac_E14, Big Bed File: /gbdb/mm10/encode3/histones/ENCFF088LWR.bigBed) by GenomicRanges::SubsetByOverlaps().

**scRNA-seq and snATAC-seq integration**. *scRNA-Seq/snATAC-Seq Integration*: For each data type, barcodes were filtered to remove far-out outliers, as previously described by Tukey's fence method. Prior to integrating data, for snATAC-Seq barcodes, gene activity score (GAS) was calculated by counting reads that mapped to the promoter regions (2 kb upstream of TSSs), the first exon of each transcript, and (if detected) presumptive enhancer loci associated to each gene. Integration of scRNA-Seq and snATAC-Seq datasets was performed using the Seurat integration workflow. GAS (snATAC-Seq) and RNA (scRNA-Seq) count matrices were normalized with Seurat::NormalizeData(), highly variable genes detected with Seurat::FindVariableFeatures() and common features across all samples selected with Seurat::SelectIntegrationFeatures(). Integration of assays was performed with Seurat::FindIntegrationAnchors() and IntegrateData().

*degPatterns*: To obtain a list of differentially expressed genes (DEGs), Seurat::FindMarkers() was used to detect DEGs from embryonic scRNA-Seq data. Barcodes were ordered by increasing pseudotime, as assigned by Monocle3. The Seurat object containing integrated snATAC-Seq/scRNA-Seq datasets was subset into two Seurat objects, one containing scRNA-Seq and one containing snATAC-Seq. degPatterns were quantified using the previously detected DEGs list and three separate matrices: gene-by-barcode counts (from scRNA-Seq Seurat object), gene-by-barcode counts (GAS slot from snATAC-Seq Seurat object) and gene-by-barcode counts (enhancer slot from snATAC-Seq Seurat object) by DEGReport::degPatterns() using the DEGReport (https://github.com/lpantano/DEGreport) package.

*RNA/GAS/Enhancers Heatmaps*: The top 500 highly variable genes from embryonic scRNA-Seq data were detected by Seurat::FindVariableFeatures(nfeatures = 500). Barcodes in the Seurat object containing integrated snATAC-Seq/scRNA-Seq were ordered by increasing pseudotime, as assigned by Monocle3. This Seurat object was subset into two Seurat objects, one containing scRNA-Seq and one containing snATAC-Seq, which were further split into three matrices: gene-by-barcode counts (from scRNA-Seq Seurat object), gene-by-barcode counts (GAS slot from snATAC-Seq Seurat object) and gene-by-barcode counts (enhancer slot from snATAC-Seq Seurat object). RNA, GAS, and enhancer gene-by-barcode matrices were concatenated together, and hierarchical clustering of genes was performed using correlation distance and

average linkage metrics. Following clustering, the concatenated matrix was re-split into RNA, GAS, and enhancer gene-by-barcode matrices. Matrices were centered and scaled for each gene prior to rendering heatmaps and the gene dendrograms were constructed using the distances calculated with the concatenated matrix. Manually annotated color bars were based on gene cohorts detected by degPatterns(), whereby the horizontal line in each box represents the median, the bottom and top edges represent the first and third quartiles, and the upper and lower whiskers extend from the edges of the box to no further than 1.5x of the inter-quartile range.

**CUT&Tag analysis**. Reads were aligned to the mouse genome (GRCm38/mm10 build) with the Bowtie2 aligner with the following parameters (-p 40 -N 1 --local --very-sensitive-local --no-unal --no-mixed --no-discordant --phred33 -I 10 -X 700), aligned reads mapping to blacklisted regions were removed, and PCR duplicates were removed with PICARD. Quality control for replicates was assessed using deepTools plotHeatmap and plotProfile, and reproducibility was assessed by peak calling per replicate with MACS and calculating pair-wise consensus peak counts with bedtools intersect using the parameter (-f 0.50). Replicates were merged with samtools merge, de-duplicated with PICARD, and peaks called with MACS using the parameter (--broad). To generate signal tracks for visualization, the merged bam files were converted to normalized bigWig files using deeptools bamCoverage with the following parameters (--normalizeUsing RPKM -p 10 --binSize 5 --minFragmentLength 150).

**Hi-C and capture-C analysis**. Hi-C and Capture-C libraries were sequenced with paired-end reads of 51 nucleotides. Data were processed using the Hi-C-Pro pipeline[63] to produce a list of valid interaction pairs. This list was converted into cool and mcool files for visualization with higlass[64]. For Capture-C data, the make_viewpoints Hicpro script was used to obtain individual Capture-C bigwig files for each replicate of each viewpoint with 2 kb-sized bins and excluding 500 bp surrounding the DpnII-fragment where probes hybridize. For visualization, averages from replicates were used.

**Statistics and reproducibility**. No statistical method was used to predetermine sample size. For the snATAC-seq experiments, we wanted a minimum of 5000 sequenced nuclei/cells per brain region from at least two replicates, which was likely sufficient to identify nearly all different cell types from each region. This goal required ~10–15,000 nuclei for each snATAC reaction (with the expectation of recovering ~30–60% of nuclei/cells for each reaction). For each timepoint and brain region, we pooled tissue from four to seven embryos, which was the amount of animals needed based on our preliminary experiments to obtain the desired amount of nuclei. Viable nuclei that passed quality control ranging from 3465–6845 nuclei/reaction (see above). For the scRNA-seq experiments, there are already several datasets in the literature we could use for comparison. Thus we were confident that one replicate of >4500 cells would be sufficient (see above). Our scRNA-seq data was in agreement with previous studies and know gene expression patterns. Per standard single-cell sequencing protocols, cells/nuclei that did not pass stringent QC measurements (% mitochondria reads, sufficient reads/cell, etc.) in the snATAC-seq and scRNA-seq datasets were considered outliers and excluded from analysis (as detailed in Supplementary Fig. 1). As stated in the Results section, we removed a 'mixed' cell population for analysis after Fig. 2. For the CUT&RUN/CUT&Tag experiments, we used 100,000 nuclei for each replicate as this amount of cells was previously optimized in our hands for these reactions. For Hi-C/Capture-C experiment, we collected 1 million cells/brain region. All computational and statistical analysis are discussed in detail above and/or in the legends of the relevant figures and tables. All attempts at replication were successful.

**General data processing**. Microsoft Excel (16.47.1) was used for data analysis and figures were generated using either Adobe Photoshop CC (20.0.9) or Adobe Illustrator CC (23.1.1)

**Reporting summary**. Further information on research design is available in the Nature Research Reporting Summary linked to this article.

## Data availability
All sequencing data (raw and processed files) generated in this study has been deposited in the Gene Expression Omnibus (GEO) database with the following accession numbers: GSE167047 (snATAC-Seq), GSE167013 (scRNA-Seq), GSE201487 (H3K4me3 CUT&Tag), GSE201488 (H3K27me3 CUT&Tag), GSE201400 (H3K27ac CUT&RUN), GSE201494 (All CUT&Tag and CUT&RUN data), GSE201186 (Hi-C), and GSE201317 (Capture-C). A searchable platform with all single-cell accessibility and transcriptomic, CUT&Tag, CUT&Run, Hi-C, and Capture-C data can be found on the UCSC Genome Browser: https://www.nichd.nih.gov/research/atNICHD/Investigators/petros/epigenome-atlas. E12.5 and E14.5 mouse forebrain H3K27ac ChIP-seq data used in this study was obtained from the ENCODE project (https://www.encodeproject.org), accession numbers

ENCSR966AIB (E12.5) and ENCSR320EEW (E14.5). VISTA enhancers information was obtained from https://enhancer.lbl.gov/. No custom code was used in the manuscript, and all computational pipelines are described in the methods. Please contact the corresponding author for more information if needed.

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

## Acknowledgements

We thank S. Henikoff for pA-Tn5 transposome; S. Coon, J. Iben, T. Li, and other members of the NICHD Molecular Genomic Core; M. Sohn for help submitting sequencing samples to the GEO database; C. J. McBain, K. Pelkey, J. A. Kassis, and members of the Petros Lab for helpful discussions. Further information and requests for resources and reagents should be directed to and will be fulfilled by the Lead Contact, Timothy J. Petros (tim.petros@nih.gov). This work was supported by *Eunice Kennedy Shriver* NICHD Intramural Awards to T.J.P., P.P.R., and R.K.D. and an NICHD Scientific Director's Award to T.J.P. and P.P.R.

## Author contributions

C.T.R and T.J.P. designed the study and wrote the paper. C.T.R., Y.Z., and T.J.P. extracted and purified nuclei. D.R.L. and T.J.P. extracted and purified embryonic cells. Y.Z. prepared single-cell sequencing libraries. C.T.R., A.M., D.R.L., and R.K.D. analyzed single-cell data. D.J.L., D.A., J.J.T., C.T.R., and P.P.R. performed and analyzed CUT&Tag and CUT&RUN experiments. J.J.T. and P.P.R. performed and analyzed Hi-C and Capture-C experiments. E.J. prepared the UCSC browser page. P.P.R., R.K.D., and T.J.P. supervised the project.

## Funding

## Competing interests

The authors declare no competing interests.
