## [Peer Review File · Nature Communications]

An epigenome atlas of neural progenitors within the embryonic mouse forebrainEditorial Note: This manuscript has been previously reviewed at another journal that is not operating a transparent peer review scheme. This document only contains reviewer comments and rebuttal letters for versions considered at *Nature Communications*.

Response to Reviewers

First, we'd like to thank the 3 reviewers for their insightful comments on our manuscript, and we believe that many of their suggestions have significantly improved the quality of our study. There are two significant changes to the resubmitted manuscript that we address directly below. Our responses to reviewer's specific comments can be found after this section, with reviewer's comments italicized and **our responses in red text**. Additionally, new text addressing reviewer's concerns are also highlighted in red in the manuscript.

1. Need for orthogonal validation and concerns about conceptual novelty

After discussion with the editor, we believe that concerns about validation and novelty were shared by all reviewers, and thus addressing these related concerns is the most important focus of our revised manuscript. To this end, we have added 3 new sets of experiments that provide a significantly greater understanding of chromatin organization in the developing mouse forebrain, validate some of our initial hypothesis from the snATAC data, and reveal novel insights in chromatin organization that are not attainable from previous published datasets using whole forebrain tissue. The new data consists of (1) region-specific histone modifications via **CUT&RUN and CUT&Tag**, (2) **region-specific Hi-C** and (3) **region-specific Capture-C** at ~50 genes with intriguing spatially-restricted expression patterns. This new data is presented in new Figure 6, Figure 7, Supplementary Table 7 and corresponding text in the Results (p. 13-15), Discussion (p. 16-18) and Methods (various sections from p. 20-34). Of note, **all of our data are now publicly available in an easily searchable and modifiable format on the UCSC Genome Browser at:**

https://genome.ucsc.edu/cgi-bin/hgTracks?hubUrl=https://hpc.nih.gov/~BSPC-Petros/track_hub/chromatin_quantification.hub.txt&hgS_loadUrlName=https://hpc.nih.gov/~BSPC-Petros/track_hub/session.txt&hgS_doLoadUrl=submit

In addition, sequencing data has been deposited at the NCBI's Gene Expression Omnibus under accession numbers GSE167047 (embryonic snATAC-Seq) and GSE167013 (embryonic scRNA-Seq) and is now publicly available, and we are currently in the process of depositing the CUT&RUN, CUT&Tag, Hi-C and Capture-C data.

By performing CUT&RUN/CUT&Tag on E12.5 MGE, LGE, CGE and CTX, we now have additional confirmation that particular loci in specific brain regions represent active promoters (H3K4me3) and repressed genes (H3K27me3). And while not conclusive on its own, H3K27ac peaks at non-promoters represent likely active enhancers, which is reinforced at many loci that line up with our snATAC peaks and Cicero connections. In addition to being one form of orthogonal validation, comparing histone modifications between different forebrain regions reveals important novel spatial heterogeneity that was not previously identified in other datasets that treat the entire forebrain as a single region.

Additionally, we performed Hi-C and Capture-C on E12.5 MGE, LGE, CGE and CTX tissue. These data reveal striking regional differences in higher-order chromatin domains and enhancer-promoter contacts, which represent novel insights in chromatin structure in developing neurons. To highlight just one example, chromatin organization at the *Nkx2-1* locus is remarkably different between the MGE and non-MGE structures (new Figure 7). To our knowledge, very few studies have performed Hi-C and/or Capture-C in neural progenitors *in vivo*, with even less comparing distinct regions of an organ that give rise to different cell types (Andrey...Mundlos *Genome Research* 2017 in the limb is the only somewhat comparable study we know of). Thus, this entire Hi-C/Capture-C dataset represents not only new and

exciting data on its own but demonstrates the feasibility of this approach to better compare/contrast higher-order chromatin organization between related but spatially distinct cell types *in vivo*.

Lastly, in addition to the factors summarized above, we'd like to emphasize an additional point as to why our dataset represents a significant advance (or more accurately, asks a different question) from previous publications and should be viewed as a critical Resource going forward. First, there are numerous studies that examine the developmental trajectory of transcriptomes and/or chromatin accessibility over time during mouse embryogenesis in whole forebrain, (e.g., Preissl...Ren *Nat Neuro* 2018; the ENCODE dataset and related publications), dorsal forebrain excitatory neurons (e.g., Telley...Jabaudon *Science* 2019; Di Bella...Arlotta *Nature* 2021) or ventral forebrain GABAergic neurons (e.g., Mayer...Satija *Nature* 2018; Bandler...Mayer *Nature* 2021). All of these studies (and ones like them) have significantly increased our understanding of genomic and epigenetic changes over time in the developing mouse embryo. However, these types of studies focus on *temporal* changes over embryonic days while our study more specifically focuses on *spatial* differences between four distinct embryonic forebrain regions that generate largely non-overlapping neuronal subtypes. While we do compare the transcriptome and chromatin accessibility between different types of neural progenitors (APs vs. BPs) and postmitotic neurons (Ns), our study does not explore the developmental trajectory of neurons over different embryonic timepoints. Thus, our study should be viewed as asking a distinct question from these previously published studies: What genetic and epigenetic differences exist between four distinct regions of the embryonic mouse forebrain that generate different, non-overlapping neuronal cell types? And we'd argue that no single study to date has provided as comprehensive a dataset to compare and contrast the transcriptome, chromatin accessibility profiles, histone modifications and higher-order chromatin structure between distinct regions of a developing organ, let alone the embryonic brain.

We hope the reviewers share this viewpoint and recognize that our improved manuscript, comprehensive dataset, and publicly searchable UCSC Genome Browser platform represent a novel, critical Resource for the field, both in our understanding of normal brain development and for future studies exploring a role for genes or SNPs in neurological and psychiatric disease.

2. Conflicting reviewer opinions on linking our dataset to human disease-associated loci

We would also like to address the reviewers' comments on linking genomic accessible regions from our dataset to disease-associated loci from the iPsych dataset (old Figure 7E-M and related text). Reviewer #1 notes that this data is '*...potentially interesting, but at this point preliminary and superficial...*' and probing the function of this region '*...would significantly elevate the impact of the manuscript.*' Reviewer #2 notes that '*...it is unclear how the authors' new data provided an advance for this analysis that was not previously possible. Can the authors compare their results to previous embryonic and adult scATAC-Seq datasets?*'. Reviewer #3 notes that we should '*...use the most recent PGC datasets that have power and therefore more interpretability (<https://www.med.unc.edu/pgc/download-results/>)*' and in general questioned our approach because '*...the datasets are generated in mouse, and non-coding regions are often not well conserved in the mouse (many did not lift over), the use of this dataset to interpret human GWAS loci is questionable. It is much more beneficial to use human defined regulatory elements for this purpose, as defined in the atlas above.*'

To summarize, Reviewer #1 wants us to expand on the particular genomic loci on Chromosome 3 we discuss in the original submission, Reviewer #2 wants us to compare this finding to other published mouse scATAC datasets, and Reviewer #3 is quite skeptical of this whole approach in general and/or wants us to use different human disease datasets. As the reviewers' opinions of this data are quite divergent, it is impossible to satisfy all of them, or honestly how best to deal with these critiques.

Regarding Reviewer #1's comments, we agree it would be interesting to probe the function of this genomic loci on Chromosome 3, but we are unaware of a clean mechanism for how to assess this function (how the SNPs affect gene expression in the human (up or down regulation) is unknown, which genes are affected is unknown, etc.). Even if we could inhibit/activate this locus in mice, the best way to evaluate these effects remains unclear. Thus, we would argue that this pursuit would be technically challenging, difficult to assess its functionality/impact, and thus beyond the scope of this manuscript.

Regarding Reviewer #2's comments, we could perform a similar analysis to other embryonic and adult mouse snATACseq (and scRNAseq) datasets, yet it's not entirely clear how this would help differentiate our study from others. It's likely that different human data sets will provide different hits (as is commonly the cases when comparing GWAS studies with different genomic loci of interest), so it's unclear how this result would add much to our findings and/or help convince the reviewer(s) about the novelty of our data.

Regarding Reviewer #3's comments, they point us to a database that contains >30 entries of ~15 neurological diseases (of which the iPsych database we used is present in this list). We could theoretically repeat the analytic pipeline we used with the iPsych database for any of these datasets, but again it's not entirely clear how this would improve our manuscript. Which datasets should we choose: SCZ, ASD, depression? That is in part why we chose a multi-disorder resource like iPsych. Performing similar analyses on multiple databases would likely result in a separate set of hits that would overlap with different genomic loci in our dataset. We could present these different set of hits in a supplementary figure, but we fear that this would still not fully satisfy the reviewers on this issue. But in general Reviewer #3 was quite skeptical of this entire approach, basically stating that one should use human defined regulatory elements for this purpose. We agree that the Liftover function has significant drawbacks due to lack of conservation between mouse and human in non-coding regions, but this strategy is often used to relate genomic regions of interest between mouse and human DNA. And better genome comparison resources are continually being developed (see Rhie... Jarvis *Nature* 2021 for example). So while Liftover might be suboptimal, it is still a useful tool in the field to compare mouse and human data.

For these reasons, we have entirely removed our analysis comparing accessible genomic loci in mice with human disease-associated loci from the revised manuscript. First, this provided the necessary space to add the new histone modification, Hi-C and Capture-C data, which we feel is more important to our study than the human analysis. Second, this change simplifies the theme and focus of our manuscript, and better positions it as a comprehensive 'Resource' characterizing the genetic and epigenetic organization of distinct neurogenic regions of the embryonic mouse forebrain. Third, we did not see a clear path forward for how best to deal with the reviewers' different concerns regarding this analysis. We plan on more thoroughly examining the relationship between neuronal subtype specific accessibility loci and relation to human disease-associated SNPs in a follow-up study dedicated to this specific question.

Reviewer #1 (NPC Genomics expert)

The study by Rhodes et al. is a single cell analysis of chromatin accessibility and its relationship with gene expression in the developing forebrain. Using snATAC-seq, they identify differential accessibility (DA peaks) that are cell type specific and change across developmental trajectories. They integrate their accessibility information with other genomic datasets (cut and tag, scRNA-seq, public ChIP-seq data) and find correlations between expression and chromatin features at developmentally-relevant genes. Neurological disorders have been associated with SNPs in enhancer regions, and the authors leverage their accessibility data to link chromatin state changes in interneuron development with these disease-associated sites.

This study is technically sound and provides a thorough description of the expression and accessibility dynamics associated with early brain development in the mouse. However, some associations (or lack thereof) remain unexplained and additional analysis is needed to address these gaps. In addition, the SNP findings remain superficial, which is a missed opportunity. Specifically:

1. Figure 2. The basal progenitors do not cluster together, yet there is not discussion or explanation for this. Are these distinct populations or is this a technical issue with the libraries?

Except for cluster 6 in Figure 2, the basal progenitor (BP) groups do largely cluster together in the hierarchical organization, each associated with their proper brain region. The first and most significant branch in the hierarchy separates the AP clusters and the BP/postmitotic neurons. 4 of the 5 BP-defined clusters (2,3,9,10) fall within this BP/N arm, with the GE BPs (2,3) adjacent to GE Ns and the Ctx/mixed BPs (9,0) adjacent to the Ctx/mixed Ns. Note that in Figure 1C, cluster 6 (defined as LGE/CGE BP) is

located adjacent to LGE/CGE AP clusters 1 and 12, as if cluster 6 could be a transitional state between APs and BPs. That could be why this BP cluster 6 fell with the AP arm in the hierarchy.

More generally, we believe that cleanly defining BPs based on their chromatin accessibility is quite challenging because of the transient nature of these cells and their continuously changing gene expression profile. APs can undergo many proliferative cycles and maintain some stable gene expression profiles (*Nestin*, *Hes* genes for example), and postmitotic neurons (Ns) are usually easy to identify based on downregulation of cell cycle genes and upregulation of some pan-neuronal genes (*Dcx*, *Dlx* genes in the GEs, etc.). In the dorsal cortex, *Tbr2/Eomes* is an accepted marker of SVZ BP cells, but *Eomes* is not expressed by GE BPs. Thus, for these reasons, we believe that cleanly defining BPs based on accessibility loci is a significant challenge (especially in the GEs lacking *Eomes*) and we are not surprised that the BP population overall could have more variability compared to the APs or Ns.

2. Figure 6. In the temporal analysis, 3330 DE genes were identified, but only 286 genes could be linked to GAS or enhancers, which is less than 10%. Does this mean that most DE does not involve accessibility changes or is this a low detection rate due to a technical reason?

The reviewer raises an interesting and important point regarding the disparity between the number of DE genes and associated enhancers. The number of genes with detectable GAS counts from snATAC-Seq is quite similar to the number of genes with detectable transcripts from scRNA-Seq (Extended Data Fig. 7e-f). The low apparent number of DE genes with enhancers is due to us using multiple methods (Cicero and SnapATAC) to characterize candidate gene-enhancer interactions. We combined enhancer predictions using 2 methods.

Cicero makes several assumptions about the nature of interactions between a gene TSS and potential enhancers. As the gene-enhancer distance increases, the Cicero model penalizes such interactions. Further, the Cicero linear model performs a LASSO step which shrinks data to a central point, effectively retaining only variables that are the most informative. The result of the Cicero model's distance penalty and the LASSO regression is that a relatively small number of gene-peak interactions will be reported, while non-informative peaks are removed. However, the final predicted gene-enhancer interactions are likely to be highly informative/reproducible for a given cell type. Indeed, when comparing Cicero enhancer predictions to known H3K27ac ChIP-Seq peaks from embryonic mouse forebrain, a high percentage of interactions intersect H3K27ac enriched regions (Extended Data Fig. 6f, 'Cicero' group).

Alternatively, SnapATAC performs a simple logistic classification, using scRNA-Seq count matrices (for all genes) as predictor variables, and binarized snATAC-Seq peaks as categorical response variables. For a given gene, the model predicts the likelihood that a gene is associated with the gene expression values in a cluster specific pattern. The process then iterates over all genes. No other regularization, penalization, or selection steps are applied to the models. Consequently, SnapATAC detects significantly more gene-enhancer interactions compared to Cicero, but many are likely to be uninformative and/or false positive interactions. This is confirmed by comparing SnapATAC enhancer predictions to known H3K27ac ChIP-Seq peaks from embryonic mouse forebrain, where only a minority of predicted interactions intersect H3K27ac enriched regions (Extended Data Fig. 6f, 'SnapATAC' group).

Since Cicero and SnapATAC emphasize different features when modeling gene-enhancer interactions, there will not be complete overlap between the model predictions (Extended Data Fig. 6f, 'Merged' group). Thus, although shared gene-enhancer interactions between the 2 methods retains a low number of common enhancers, these enhancers are more likely to be regulating transcription. We have validated several examples of promoter-enhancer interactions using orthogonal methods of genome-wide chromatin quantification (Fig. 6). There are numerous ways to quantify DE genes and associated GAS and enhancer counts that have advantages and disadvantages, but our method was designed to recover the most informative candidate enhancers to explore in future studies.

Descriptions of our Cicero and SnapATAC enhancer prediction methods are detailed in the 'snATAC-Seq Analysis' section of the Methods (p. 28-31). For clarity, we have revised the relevant text in the Results section to avoid confusion about the number of DE genes and associated enhancer predictions (p. 11, 280-282): *'As not all genes had detectable snATAC based GAS and/or enhancer counts, we*

selected ~300 of the most differentially expressed genes (DEGs) (from scRNA) among APs, BPs and Ns that had corresponding GAS and enhancer counts.'

3. Figure 7. One issue with the manuscript relates to the integration of the interneuron (p30) data with the earlier neurons. The UMAPs indicate completely distinct cell populations, and I am surprised that the embryonic neurons are not more closely related to the more mature neurons, either in expression or accessibility profiles. This section requires a deeper analysis.

While this data is no longer in the revised manuscript, we will still address this point. The reviewer is correct that the integrated embryonic and P30 cell populations form distinct, separate clusters in Old Figure 7A-D (and the snATAC and snRNA plots individually in Old Fig S7). It is actual quite common for significant separation in single cell space between embryonic and mature neurons because of the large changes in gene expression between these two timepoints. Some recent examples from the literature are below. Kim...Blackshaw *Nat Comm* 2020 shows that the P45 (dark brown) hypothalamic cells form distinct clusters that are predominantly separated from the embryonic and early postnatal timepoints. A similar clustering pattern is observed in midbrain dopaminergic cells in Tiklova...Perlmann *Nat Comm* 2019, with the P90 cells (dark brow) forming distinct clusters from the embryonic and early postnatal timepoints. Lastly, Hammond...Stevens *Immunity* 2019 paper demonstrates that microglia harvested from P100 (grey dots on left plot) are largely segregated from the embryonic and early postnatal timepoints. Thus, the significant changes in gene expression throughout development usually results in embryonic and mature cell types forming distinct clusters when integrated in the same tSNE/UMAP plots.

**Kim...Blackshaw 2020
Hypothalamus**

**Tiklova...Perlmann 2019
Midbrain Dopaminergic cells**

**Hammond...Stevens 2019
Microglia**

4. Also related to figure 7, the association with human SNPs is potentially interesting, but at this point it remains a preliminary and superficial observation. A region on chromosome 3 is identified that contains several loci, but there is no effort to probe further to better understand the biological and functional implications. This would significantly elevate the impact of the manuscript.

Please see our response in the 'Conflicting reviewer opinions on linking our dataset to human disease-associated loci' section above regarding the human disease-associated loci in old Figure 7.

Minor comments:

5. S1A and B. The labels along the x-axis are very difficult to read. These should be spread out.

We have modified the X-axis of these panels in Extended Data Fig. 1 to make them more legible.

6. Fig 4F. Should the y-axis read "# of connections"? Or "# of interactions"?

The y-axis label now reads "# of interactions (thousands)" in what is now Extended Data Fig 6F.

7. S4. Define SCT in legend.

To clarify this issue, the legend of Extended Data Fig. 4 now reads 'RNA' and 'GAS'. SCT did stand for Single Cell Transform normalized RNA counts (SCTransform()) in Seurat).

Reviewer #2 (Genetic regulation of cortical development expert)

In this Resource, Rhodes and colleagues sought to characterize the chromatin landscapes associated with the developmental trajectories of forebrain neuronal subtypes. The authors generated and integrated single-nucleus chromatin accessibility and single-cell transcriptome data from regions of the developing forebrain. They used snATAC-Seq to identify differentially accessible regions and intersected these regions with scRNA-Seq data (to examine accessibility versus transcript abundance) and the iPSYCH2012 dataset (to examine correlations between disease-associated polymorphisms and accessible regions). There are however several issues that limit the significance and utility of this Resource.

1. The first concern is overlap and advance over published works. The opening sentence of manuscript, "... a detailed characterization of chromatin accessibility during neurogenesis has not been explored.", is an unfair description of the field. Of particular importance, Preissl et al., Nat Neurosci, 2018, performed snATAC-Seq on forebrain nuclei at seven developmental ages: E11.5, 12.5, 13.5, 14.5, 15.5, 16.5, and P0. Here, although Rhodes et al. microdissected the GEs, this analysis was carried out only at a single age (E12.5). The lack of comprehensive longitudinal analysis limits tracking of chromatin changes across development and the study of developmental trajectories as stated in the manuscript title. It also reduces the additional utility and advance of this study over published works.

We apologize to the reviewer for our title being misleading, for inaccurately representing the state of the field in our opening sentence, and for not defining what we meant by 'developmental trajectories' (AP to BP to postmitotic neurons) in our manuscript. To more accurately convey the theme of our study, we have changed the title to 'An epigenome atlas of neural progenitors within the embryonic mouse forebrain'. And we have altered the language in our manuscript to clarify this point and eliminate the implication that we are studying developmental trajectory across different timepoints.

We hope that our new datasets including histone modifications, Hi-C and Capture-C in the E12.5 MGE, LGE, CGE and CTX (and related analysis therein) highlight the significant advance of our study of published works. We emphasize both the novelty of our resubmitted manuscript and how it should be viewed in comparison to other studies in the 'Need for orthogonal validation and concerns about conceptual novelty' above. But briefly, our study is more focused on comparing spatial regions giving rise to distinct neuronal subtypes rather than tracking accessibility changes across different developmental timepoints like many published works.

We view the Preissl...Ren *Nat Neuro* paper as an important study that provides significant insight into our understanding of chromatin accessibility changes in the forebrain throughout neurogenesis (E11.5-P0). Especially being published in 2018, this study was at the forefront of performing snATAC experiments *in vivo* and should be regarded (and cited) as such. However, we highlight several significant differences between our study and Preissl et al., several of which the reviewer already notes. First, Preissl used entire forebrain whereas we microdissected distinct forebrain regions. This provided us with the ability to definitively compare and contrast ATAC peaks between different brain regions, which cannot be performed with the Preissl dataset. Second, Preissl et al., performed scATAC-seq on ~12.7K cells from 7 different timepoints ranging from E11.5 to P0, with 2 replicates/timepoint, which averages out to ~900 cells per replicate (or ~1800/timepoint, see Sup Table 2 in their paper), plus an additional ~3,000 cells from P56 forebrain. Comparatively, our snATAC dataset consists of ~8,500-11,300 nuclei per brain region (2 replicates/region). This increase in cell numbers gives our study significantly more power to identify differentially accessible peaks between different forebrain regions *and* between different neuronal progenitor cell types (APs vs. BPs. vs. Ns) within specific regions. As we found in a different study from the Petros lab (<https://www.biorxiv.org/content/10.1101/2021.07.05.451224v1>), increasing cell number (and thus power) can reveal completely novel gene expression patterns (or in this case accessibility profiles) that were not apparent in previous publications using fewer cells and/or collecting whole forebrain as compared to harvesting tissue from distinct forebrain regions.

2. The authors carried out snATAC-Seq and scRNA-Seq on different samples. Based on snATAC-Seq data, a Gene Activity Score (GAS) was calculated as a proxy for transcript abundance such that the

snATAC-Seq and scRNA-Seq data could be integrated for clustering. This integrated cell clustering was dependent on concordance between snATAC-Seq and scRNA-Seq data. The resulting cell clusters were then used to interrogate the relationship between gene accessibility (snATAC-Seq) and expression (scRNA-Seq). This is a circular analysis. If the biological truth is that single cells can exhibit discordant snATAC-Seq and scRNA-Seq, such cells would be missed by this analysis as it is based on concordance. The authors' assertion that, "Transcription, gene accessibility and active enhancer utilization are highly correlated", therefore carries a serious confound.

The reviewer is certainly correct that the integration analysis is influenced by, and dependent upon, concordance between snATAC-Seq and scRNA-Seq data. However, a recent paper (Allaway... Fishell, *Nature* 2021) utilized single cell ATAC-Seq, single cell RNA-Seq and 10X Genomics Multiome analysis in a longitudinal study exploring the genetic and epigenetic dynamics in a single family of developing interneurons. Among other findings, Allaway et al. support the notion that when measured snATAC-Seq and scRNA-Seq from the same cell, critical cell-type markers have extremely high concordance. We are using the 10X Genomics Multiome kit for other ongoing experiments in the lab and our preliminary analysis supports the findings by Allaway et al. Other groups have reported similar concordance between single assay snATAC-Seq and single assay scRNA-Seq data sets, both in terminally differentiated cell types as well as developing cells (e.g., Zhang...Ren *Cell* 2021; Li...Ren *Nature* 2021; Domcke...Shendure *Science* 2020. Throughout the manuscript, text has also been reworded to avoid confounding/misleading statements such as "Transcription, gene accessibility and active enhancer utilization are highly correlated".

Additionally, the reviewer's critique is most important for temporally regulated genes where there would likely be a (slight?) temporal mismatch between chromatin accessibility and gene expression. However, as we have emphasized in the revised manuscript, our focus is primarily on differential transcriptome and epigenome state between distinct regions of the embryonic mouse brain. For spatially restricted/enriched genes, there is a high concordance between genes and GAS (as demonstrated in Fig. 1, Fig. 4e-j, and Extended Data Figs. 1, 2 and 5). In fact, tissue specific markers tend to be the genes that hold the most weight with Seurat integrations.

3. In addition to the above issue, multiomic studies on single cells have become broadly available in the past year. Most notably, the 10x Multiome analysis enables snATAC-Seq and scRNA-Seq on the same cells, enabling accurate cell clustering based on expression and direct data integration without the confounds of integrating snATAC-Seq and scRNA-Seq from different samples. Unfortunately, this Rhodes et al. dataset may become obsolete in the not too distant future.

We are aware of the 10X Genomics Multiome kit and are currently using it for other ongoing experiments in the lab. This kit was released in September 2020 after we had generated most of the snATAC and scRNA data that is in this manuscript. We agree that this kit does eliminate some of the confounds related to integrating snATAC and scRNA datasets and it will be a critical tool going forward.

However, we would like to push back on the notion that our dataset '...may become obsolete in the not too distant future.'. New strategies to improve the quality, efficiency and multidimensional analysis of single cell sequencing techniques are being published every week (let alone the advancement of spatial transcriptomics), and these advances will continue to increase the power that researchers have at their disposal. The idea that these advances make previous datasets that utilize earlier technology 'obsolete' seems shortsighted. We (and plenty of other researchers) still explore published datasets that may not employ the latest state-of-the-art technology to identify candidate genes or genomic loci for our particular interests. Furthermore, scATAC-seq technology has been around since 2015 (Buenrostro...Greenleaf *Nature* 2015), and a commercially available scATAC-seq kit from 10X Genomics was released in November 2018. Since that time, no one has generated a DNA accessibility map of spatially distinct neurogenic regions in the embryonic mouse brain like we have here. So the idea that another lab is (or will be soon) generating a Multiome dataset in the same manner as we did here, while not impossible, seems unlikely. We believe it's more likely that other labs may perform Multiome experiments on different embryonic timepoints (as in Allaway...Fishell *Nature* 2021) rather than distinct spatial forebrain regions, Furthermore, comparing and contrasting similar datasets (either scRNA, scATAC, or multiome)

generated by different labs is critical for the community to ensure the quality of individual datasets; the more datasets that are in general agreement, the more reliable and accurate they are. Thus, we are quite confident that our snATAC dataset (especially with the addition of histone modifications and Hi-C/Capture-C) will not become obsolete anytime soon.

4. For the snATAC-Seq analysis, the authors pooled 4-7 embryos and sequenced 2 replicates totaling ~5000 nuclei. These numbers are low in comparison to recent studies in which each replicate often consists of >10k nuclei. The nature of the replicates (technical/biological) were not described. A concern over pooling is that inter-individual variations would be completely masked. Whereas as an n=2 individual animals is the absolute minimum for any analysis, pooling several animals together for n=2 pools averages out variations and can create the false impression of consistency. For the scRNA-Seq analysis, the authors sequenced n=1 to >4000 nuclei. For a Resource paper that depends on the strength of the sequencing data, this is far from ideal.

The reviewer is correct that we performed 2 snATAC-seq replicates for each brain region (MGE/LGE/CGE/Ctx) that resulted in ~8,500-11,300 nuclei per region in total. And we also performed 2 snATACseq replicates for the P30 *Dlx6aCre;Ai9* cortex and hippocampus resulting in ~6,500 and 8,750 cells, respectively, although this data has been removed from the revised manuscript.

However, we disagree with the notion that our replicates and cell numbers are low in comparison to recent single cells sequencing studies. First, in using the droplet based 10X Genomics platform, 10X recommends ~10-12K cells/nuclei per reaction with an expected recover of 50% (~5-6K cells). If you go higher than 10-12K cells, there will be a significant increase in doublets that can need to be removed and this can complicate analysis. Our results are consistent with their recommendations, as we usually recovered ~35-60% of the expected cell numbers in our reactions. More recent strategies have been developed with additional barcoding steps that can significantly increase the number of cells sequenced in 1 reaction (for example, Lareau...Buenrostro *Nat Biotech* 2019; Shin...Bang *Science* 2019, among others), but most standard droplet-based/10X Genomics experiments recover 4-6K cells per reaction, not >10K as suggested by the reviewer.

For comparison to other recent droplet-based single cell sequencing papers of embryonic mouse forebrain: (1) Di Bella...Arlotta 2021 *Nature* paper performed a comprehensive scRNAseq analysis of mouse embryonic dorsal cortex every day from E10.5-E17.5. This study consisted of 1 replicate on all cortical embryonic timepoints from E10.5-E17.5, with a range of ~3,000-11,600 cells per timepoint (mean ~7,000/timepoint). (2) Moreau...Causeret *Development* 2021 paper analyzed a total of 4,225 cells from 1 replicate near the pallial/subpallial boundary. (3) Telley...Jabaudon *Science* 2019 paper analyzed a total of 2,756 cortical cells spread across 12 different embryonic cell labeling and collection conditions, 1 replicate each, which averages out to ~230 cells/timepoint. (4) Preissel...Ren *Nat Neuro* 2018 performed scATAC-seq on ~12.7K cells from 7 different timepoints ranging from E11.5 to P0, with 2 replicates/timepoint, which averages out to ~900 cells per replicate (or ~1800/timepoint, see Sup Table 2 in their paper). (5) Mi...Marin *Nature* 2018 paper sequenced a total of 2,000 cells from 3 different brain regions at 2 different embryonic timepoints (1 replicate each), which averages out to ~330 cells/timepoint. (6) Mayer...Fishell *Nature* 2018 paper is probably closest to the standards proposed by this reviewer (and our standards as well), performing scRNAseq on a total of 5,600-8,500 cells from the MGE, LGE and CGE, with multiple replicates of each region.

There certainly are other relevant single cell sequencing papers in the literature, but we think this cohort of papers (all of which are high-quality papers published in well-respected journals, most of which are cited in our manuscript) accurately reflects the state of broadly acceptable parameters for cell/replicate number. We would argue that our dataset clearly equals, and more often exceeds, the total number of replicates and cell number per timepoint/condition compared to these other manuscripts. Thus, we disagree the reviewer's characterization of how the replicates and cell numbers in our snATAC-seq dataset relates to other comparable single cell sequencing papers.

The reviewer correctly notes that we only performed 1 replicate of scRNA-seq reactions for each embryonic brain region. This was due in part to the other published data sets that had performed scRNA-

seq in both the GEs of the embryonic mouse brain (Mi...Marin *Science* 2018; Mayer...Fishell *Nature* 2018). Since our data was in agreement with these previous datasets, and the main focus of our study is on the epigenomic organization, we did not feel the need to repeat the scRNA-seq experiments. As noted in the previous paragraph, many well-respected single cell sequencing experiments only performed 1 replicate for their data.

5. *Another experimental concern is the lack of orthogonal validation of the data, predictions, or discoveries. For example, the authors highlighted the discovery of cis-regulatory elements and 'high-confidence' enhancers. Experimental validation of a few of these putative regulatory regions (for example by luciferase assay) would enable calculation of a false positive rate of the predictions and add confidence to the validity of the rest of the untested predictions. In addition, the authors detected some previously validated VISTA enhancers. Of these known enhancers, how many were unbiasedly predicted by the authors' own data? How many were missed? It would be valuable to establish a false negative rate to add statistical rigor to the predictions.*

We thank the reviewer for this comment and agree that experimental validation of regulatory elements can increase confidence in cRE predictions. The additional experiments involving histone modifications, Hi-C and Capture-C in each brain region represents orthogonal validation of the snATAC data (and in fact significantly extends our understanding of potential spatially-restricted chromatin organization).

One important finding is that every element of our dataset (transcriptome, accessibility profile, histone modification, higher-order chromatin organization) is critically dependent on the brain region and/or cell type (AP vs. BP vs. postmitotic neuron). The luciferase assay is a useful tool to test candidate enhancers in an artificial *in vitro* system. But we feel that losing the context of cell type and chromatin organization in a luciferase assay reduces the utility of extrapolating these results to forebrain neural progenitors. Especially in light of our new histone modification and Hi-C/Capture-C data showing striking changes in chromatin organization between different brain regions that cannot be recapitulated in an artificial system. Instead, we are currently establishing a CRISPR-based system to activate or repress genes (or candidate cREs) both *in vivo* and in mESC differentiations *in vitro* using 2 recently published transgenic mouse lines (Gemberling...Gersbach *Nat Methods* 2021). Our future studies will be able to more directly assess cREs and promoter-enhancer function while retaining cell type and region-specific epigenomic organization.

Regarding the overlap with VISTA enhancers, this is a difficult question to answer accurately and concisely for multiple reasons. First, the current VISTA database (<https://enhancer.lbl.gov>) consists of 3233 tested elements of which ~50% (1654) display some sort of expression profile in the E11.5 mouse. Second, VISTA calls any expression in the embryo 'validated' regardless of whether the expression pattern aligns with some known gene (ideally a gene near the candidate VISTA enhancer). Thus, it is extremely challenging to look at VISTA hits/expression alone and determine which genes correlate with the expression profile in the VISTA mouse. For example, if you type *Nr2f1* into the VISTA search browser, it returns 31 VISTA hits near the mouse *Nr2f1* gene, of which 17 show some expression in the E11.5 mouse. However, the expression profile of these 17 hits are all over the map; some are enriched in various regions of the brain (forebrain, midbrain, hindbrain) and spinal cord, others in the nose, eye, limbs and/or facial mesenchyme. And some lines depict significant expression variability even between littermates. One would have to look at the expression profile of all genes nearby these VISTA hits (if the gene expression profile is even known) to try to make logical sense of these expression patterns. We could certainly look at co-accessibility profiles and Cicero connections in our snATAC data to see how many of these VISTA hits connect to the *Nr2f1* promoter (or any other gene of interest). But it's not clear how useful this type of analysis is because of the lack of clarity on how VISTA expression patterns relate to the gene of interest. And the concept of 'validated' VISTA hits being 'missed' in our analysis may actually be a good thing if the VISTA hit displays an expression profile distinct from the gene of interest. Thus, for these reasons, we do not believe a comparison between our snATAC/Cicero connections and validated VISTA hits would provide useful analysis.

6. *The authors intersected their data with the iPSYCH2012 dataset to link accessible genomic regions with psychiatric disorders. I am unable to evaluate the statistical rigor of this analysis because the authors*

sub-divided the gene lists multiple times, and showed overlap without presenting overrepresentation or hypergeometric tests. It is unclear if multiple testing has been corrected for. The presentation of the work is confusing. The Venn diagram in Fig. 7F seems to suggest that there were only 73 DEGs total from the dataset. If this is not the case, then the diagram is incorrect. Based on the boxplots in Fig. 7H (which had unlabeled annotations), it would appear that the median odds ratio was below 1.0. How are these variants associated with disease? Importantly, it is unclear how the authors' new data provided an advance for this analysis that was not previously possible. Can the authors compare their results to previous embryonic and adult scATAC-Seq datasets? Showing, for example, that previous datasets are insufficient for embryonic cREs (Preissl et al., Nat Neurosci, 2018) or mature interneuron cREs (Fang et al., Nat Commun, 2021) could support that this study provides an advance over previous works.

Please see our response in the 'Conflicting reviewer opinions on linking our dataset to human disease-associated loci' section above regarding the human disease-associated loci in old Figure 7.

7. The value of a Resource depends on the data being accessible to a wide audience. Previous Resource papers (e.g. Pombo Antunes et al., Nat Neurosci, 2021, Blum et al., Nat Neurosci, 2021, Eze, et al., Nat Neurosci, 2021) were accompanied by an interactive tool that provides public access to the data. Providing such a tool (in addition to raw data deposited to GEO) would greatly increase the accessibility and potential utility of this Resource.

We thank the reviewer for this suggestion and we agree that making our dataset more easily accessible will significantly increase its utility to a broader audience. In addition to making the data publicly available via GEO deposits, we have utilized the UCSC browser to integrate all of our data into one searchable and modifiable platform for the general public using the UCSC Genome Browser:

https://genome.ucsc.edu/cgi-bin/hgTracks?hubUrl=https://hpc.nih.gov/~BSPC-Petros/track_hub/chromatin_quantification.hub.txt&hgS_loadUrlName=https://hpc.nih.gov/~BSPC-Petros/track_hub/session.txt&hgS_doLoadUrl=submit

Individual users can enter a gene or genomic coordinates and have instant access to our scRNA, snATAC, Cicero connections, Cut&Tag/CUT&RUN, Hi-C and Capture-C data in one window. And users have the flexibility to rearrange, remove and recolor any track or group of tracks using standard UCSC Genome Browser controls. Users can also download any/all of the data hosted on our UCSC Genome Browser track hub, so they can incorporate our data when analyzing their own experimental data. We are confident that this addition will significantly increase accessibility and utility of our dataset to the broader scientific community.

Reviewer #3 (NPC Genomics expert)

The authors generated snATAC-seq data from ~10K cells in each of LGE, CGE, MGE, and cortex of the developing mouse brain. They also generated snATAC-seq/scRNA-seq from sorted and labeled inhibitory neurons of the adult mouse brain. Then, the authors assigned peaks with differential chromatin accessibility across cell types, mapping those non-coding regulatory elements to genes of action using peak-peak correlation or mapping peaks to genes with a combination of scRNA-seq and scATAC-seq, and finally utilizing the iPSYCH 2012 GWAS dataset together with this developmental gene regulatory data to help understand cell types involved in risk for multiple neuropsychiatric disorders.

The results from these analyses were largely confirmatory and novel regulatory elements had no experimental validation. While the focus on inhibitory neurogenesis is interesting, there are large scATAC-seq datasets already existing in developing mouse brain ([https://www.cell.com/cell/fulltext/S0092-8674\(18\)30855-9](https://www.cell.com/cell/fulltext/S0092-8674(18)30855-9)) and human fetal brain (<https://science.sciencemag.org/content/370/6518/eaba7612?rss%253D1>) and adult human brain (<https://www.nature.com/articles/s41588-020-00721-x>) that I believe have already identified cell types and regulatory elements in the developing/adult brain, and mapped them to GWAS loci. Because of this, unfortunately I did not find the study very novel or providing new insights.

The first study cited by the reviewer is Cusanovich...Shendure 2018 *Cell* paper that performed scATAC-seq on adult mouse tissue, not developing/embryonic mouse brains. So, this Cusanovich reference is not relevant to our dataset in this context. It's possible the reviewer meant to refer to the Preissl...Ren *Nat Neuro* 2018 paper that performed snATACseq on the embryonic mouse brain. We discuss our manuscript in respect to this Preissl paper in our response to Reviewer #2, point #1 above.

Regarding the reviewer's concern about the novelty/new insights of our study, please see our response to this issue in the 'Need for orthogonal validation and concerns about conceptual novelty' section at the start of this Response. But briefly, we believe that the new additions of the histone modification, Hi-C and Capture-C do provide exciting new insights into potential epigenetic regulation of gene expression. Relying on only 1 modality to identify putative enhancers (as with snATAC in our original submission) is likely to overlook many genomic loci of interest. The additional experimental data both support previous predictions (from our data and publicly available sources such as VISTA) and generates new cREs that have not been previously explored. For example, we identified an MGE-specific genomic interaction between the *Nkx2-1* promoter and the *Mbip* gene, and non-MGE specific interactions between the *Nkx2-1* promoter and both the *Nkx2-9* and *Pax6* regions. This highlights a previously unknown region-specific chromatin organization that likely has important roles in how these genes are expressed or repressed in particular brain regions. In scanning spatially-restricted genes, we find many examples where our ATAC co-accessibility, K27ac hits and/or Hi-C/Capture-C data are aligned with previously identified enhancers (either from VISTA or other sources), and many loci of putative promoter-enhancer interactions that have not been described before. Thus, in particular with the new data, we believe that our study does highlight numerous novel insights of chromatin organization in specific regions of the mouse embryonic forebrain.

I have some specific comments as well:

1. *The authors use iPSYCH2012 dataset to assign disease relevance to genomic loci, and interpret those loci through the dataset generated here. The authors use a mapping of "disease associated genes" which mapped non-coding variation to (presumably?) closest gene. This should be better described, especially because closest gene mapping is known to be inaccurate about 50% of the time. Additionally, this dataset has relatively few genome-wide significant SNPs. . I would encourage the authors to use the most recent PGC datasets which have higher power and therefore more interpretability (<https://www.med.unc.edu/pgc/download-results/>). The specific SNPs selected for overlap with peaks identified here did not survive genome-wide significance and therefore cannot be reliably associated with neuropsychiatric disorders.*

Please see our response in the 'Conflicting reviewer opinions on linking our dataset to human disease-associated loci' section above regarding the human disease-associated loci in old Figure 7.

2. *Because the datasets are generated in mouse, and non-coding regions are often not well conserved in the mouse (many did not lift over), the use of this dataset to interpret human GWAS loci is questionable. It is much more beneficial to use human defined regulatory elements for this purpose, as defined in the atlas above.*

Please see our response in the 'Conflicting reviewer opinions on linking our dataset to human disease-associated loci' section above regarding the human disease-associated loci in old Figure 7.

3. *Batch effects/randomization/batch effect removal were not described in the methods. One can see that TSS enrichment was lower in *cge/lge/mge/cortex1* as compared to 2 suggesting that multiple rounds of library preparation in 10X yielded different quality results (Supplementary Figure 1). It would be helpful to have a better description of how this was corrected for in downstream analyses.*

The reviewer is astute to notice that there were slight differences in TSS enrichment between the first and second replicates. Such differences are relatively common when preparing multiple batches of single cell libraries, necessitating a consistent strategy to address batch variation. We used similar but distinct strategies to address batch variation when analyzing snATAC-Seq only compared to integrating scRNA-

Seq and snATAC-Seq datasets. We used the integration framework developed by for the Seurat and Signac single cell analysis packages (Stuart ... Satija, *Cell* 2019; Stuart ... Satija, *Nature Methods* 2021).

Batch effect strategy for ATAC data:

```
FindTopFeatures()  
RunTFIDF(pbmc.combined)  
FindIntegrationAnchors()  
IntegrateEmbeddings()
```

More generally, the process is as follows:

1. Detection of highly variable features common to all batches. In our case, these are peaks common to all samples.
2. Construct a weights matrix that defines the association between cells in one batch (reference) and another batch (query) using low dimensional cell embeddings (LSI coordinates).
3. Compute a transformed dimensional reduction matrix and merge the Seurat objects.

This is the preferred method for incorporating single cell ATAC-Seq data sets which tend to be extremely sparse. The resulting dimensional reduction has been shown to remove technical variation seen across batches at the low dimensional level and it does not overcorrect between batches, which can occur using other methods such as `Seurat::IntegrateData`. Using an example visualization from the Signac website (https://satijalab.org/signac/articles/integrate_atac.html) we can see efficient incorporation of cell embeddings in reduced dimensional space:

Importantly, because the peak-by-barcode count matrices are not transformed in any way (e.g., TSS enrichment scores are not multiplied by a weights vector), the Signac and Seurat authors have provided built-in functionalities for all downstream analysis steps to regress out confounding variables across batches using a linear model. Effectively, the batch correction is done “on-the-fly” by all methods within Signac/Seurat workflows.

Batch effect strategy for RNA/ATAC integration:

```
NormalizeData()  
FindVariableFeatures()  
SelectIntegrationFeatures()  
FindIntegrationAnchors()  
IntegrateData()
```

More generally, the process is as follows:

4. Detection of highly variable features common to all batches. In our case, these are peaks common to all samples.
5. Construct a weights matrix that defines the association between cells in one batch (reference) and another batch (query).
6. Compute the integration matrix for highly variable features (from step 1 above).
7. Compute the transformation matrix as the product of the integration matrix (from step 3 above) and the weights matrix (from step 2 above).
8. Subtract the transformation matrix from the original expression matrix.

During the standard integration process, high dimensional cell-by-feature count matrices are transformed into a normalized, batch corrected matrix. The process list above is outlined in the Seurat user manual, under the IntegrateData() documentation (<https://satijalab.org/seurat/reference/integratedata>) and works well for an arbitrarily large number of batches.

For clarity, we have emphasized our batch removal strategies for both usage cases (ATAC only and RNA/ATAC) in the relevant Methods sections on p.28 lines 715-721 and p.31 lines 791-796.

4. *Abstract and intro do not describe that these experiments were done in a mouse.*

The second sentence of the Summary from the original (and current) submission states that ‘We collected single-cell chromatin accessibility profiles from four distinct neurogenic regions of the embryonic mouse forebrain using single nuclei ATAC-Seq (snATAC-Seq).’ In the introduction, we added ‘...we characterized the chromatin accessibility of cells during the transition from progenitors to lineage restricted neurons within the GEs and dorsal telencephalon of the embryonic mouse brain.’ (p.4 lines 77-80). And the new title now reads ‘An epigenome atlas of neural progenitors within the embryonic mouse forebrain’.

5. *Fig 1E Pseudotime spelled incorrectly*

Pseudotime now spelled correctly in Fig 1E.

6. *Referring to neurons as postmitotic neuronal precursor (N) was confusing to me. Precursors are usually used to name a type of progenitor cell rather than a neuron.*

We understand this point as we also struggled with how best to define these cells since there is not a universally accepted nomenclature for this newly postmitotic/immature neuron. But we agree that the term ‘neuronal precursor’ is confusing in this context and have instead renamed these cells ‘postmitotic immature neurons (N)’ to more accurately define this cell population throughout the text.

7. *Why were the three non-neuronal clusters removed? “of 94 which three non-neuronal clusters were removed based to retain 96.8% of nuclei in 24 clusters”*

We wanted to restrict all of our analysis to neural progenitors and neurons that were derived from the 4 regions of interest. At E12.5, these regions also contain cells of the vasculature system (endothelial cells, pericytes, etc.), ependymal cells that line the lateral ventricle, and possibly other non-neuronal cell types. A similar filtering strategy is often performed in other papers (e.g., Mayer ... Satija, *Nature* 2018). Figure S10-Q display the cell clusters that were removed for further analysis. A modified version of Fig S1Q is below, which shows the 3 clusters that were removed from all analysis. Cluster #26 likely consists of astrocytes and/or microglia based on strong expression of APOE. Clusters #20 and 22 were more difficult to define based on gene expression, they could be stressed/dying cells or doublets.

8. *“We detected a total of 30,046 DA peaks (FDR <= 0.05, average log(fold change) > 0)” why only using fold change >0 peaks (peaks opening in response to different cell types?)*

The aim of this analysis was to detect cell-type specific markers at the chromatin accessibility level. Along similar lines, one can perform differential gene expression analysis to detect A) cell type biomarkers (only cluster-specific upregulated genes) or B) all differentially expressed genes (upregulated or downregulated) to characterize changes between groups.

To explore these applications further using AP-specific gene *Nes* as an example, we can look for clusters in which *Nes* is differentially upregulated ($\log(\text{fold change}) > 0$), which is indicative of *Nes*⁺ progenitors (case A). If the goal is to find *Nes*⁺ populations, looking for clusters where *Nes* was differentially downregulated ($\log(\text{fold change}) < 0$) would not be as informative since many cell types can be *Nes*⁻ (BPs, postmitotic immature neurons, mature neurons and most non-neuronal cells).

Similarly, a central goal of using snATAC-Seq data was to detect cis-regulatory elements specific to a population, regardless of how those DA peaks may change (opening vs closing) during development. To avoid confusion between these two usage cases for differential analysis, we have included an explanatory statement in the revised manuscript on p. 6, lines 142-143: *These DA peaks represent accessible genomic loci that are potentially unique to specific cell types.*

9. I'm having difficulty understanding the reasoning behind the "mixed" population that could be from contamination of CGE/LGE with cortex. If these "mixed" cells were derived from cortex, why are they not just matching with the cortical cluster?

After further investigation, we are confident that this 'mixed' population from the LGE/CGE with accessibility at excitatory-associated genes arises from cells at the pallial-subpallial boundary (PSB). As noted in a recent paper that performed scRNAseq on this area (Moreau...Causeret *Development* 2021 and references therein), this PSB region is a 'fuzzy' border between the dorsal and ventral telencephalon that becomes more refined over time. This lateral/ventral pallium area gives rise to cells constituting the piriform cortex, claustrum and amygdala. There are no sharp molecular boundaries at this region, and instead there are graded gene expression patterns. In fact, *Pax6* expression does not end abruptly at the physical PSB, but instead it extends into the anatomical LGE and CGE (see E12.5 *Pax6 in situ* below from Allen Brain Atlas, red arrow indicates physical PSB). Additionally, some of the *Pax6*-derived cells can be seen migrating through the deeper portion (mantle) of the LGE (green arrows) and CGE (not shown). When performing the dissections, we use the physical PSB boundary and take the LGE region demarcated by the blue lines in the figure. We did the dissection this way to ensure that we obtained cells from the most dorsal and ventral LGE/CGE regions to maximize our understanding of the heterogeneity in these regions. However, it appears that we likely collected some of the *Pax6*⁺ PSB-derived cells in the process. And since these cells populate the claustrum and amygdala, it's likely that their gene expression profile (and thus accessibility patterns) differs from the dorsal cortex that generates purely Layer II-VI cortical excitatory cells. This also explains why we do not observe these cells from the MGE dissection, as these *Pax6* excitatory cells do not extend ventrally into the MGE.

We have added the following sentences to the text to better explain the origin of these cells, p. 5 line 120-122: *This was likely contamination from the pallial-subpallial boundary (PSB), a region that gives rise to cells located in the piriform cortex, claustrum and amygdala^{26,27}.*

And we have fleshed this point out more in the Discussion, p.17-18 lines 432-439: *'Third, the population of 'mixed' cells that were collected with the LGE and CGE tissue expressed markers for both GABAergic and glutamatergic cells yet formed a distinct cluster from the cortex and GE populations (Fig. 2 and Extended Data Fig. 2). These 'mixed' cells likely reside at the PSB as they were not detected in the MGE population. The diversity of cells arising from the lateral/ventral pallium remains poorly characterized, but this region appears to give rise to glutamatergic cells of the insular cortex, piriform cortex, claustrum and pallial amygdala²⁶. A recent scRNAseq study may shed light on the heterogeneity of this PSB region²⁷, but further work is needed to better characterize this cell population.'*

REVIEWERS' COMMENTS

Reviewer #1 (Remarks to the Author):

My comments have been adequately addressed and I support publication in Nature Communications. My only suggestion for the authors is to update the references section, as some of the bioRxiv preprints may have been published in peer reviewed journals at this point.

Reviewer #2 (Remarks to the Author):

In this revised manuscript, Rhodes et al added a substantive amount of new sequencing data (e.g. region-specific CUT&RUN and CUT&Tag data of H3K4me3, H3K27me3, and H3K27ac, region-specific Hi-C chromatin conformation data, and region-specific Capture-C data targeting 50 genes). These new data validated and extended their original snATAC-seq data, and provided new insights into region-specific chromatin organization in the developing forebrain. Importantly, they have made the data available and easily accessible to the public via UCSC genome browser tracks. They have also removed the comparison with the human iPSYCH2012 disease-associated loci dataset, updated the title, and made appropriate textual changes. My previous concerns have been addressed. I commend the authors for these efforts.

Reviewer #3 (Remarks to the Author):

I find the revised manuscript to be much improved and very interesting. I am especially interested in the region specific enhancer activity shown in the last figure. I support publication of this manuscript.